# Cytoneme-mediated intercellular signaling in keratinocytes is essential for epidermal remodeling in zebrafish

Yi Wang[1], Thomas Nguyen[1], Qingan He[1], Oliver Has[1], Kiarash Forouzesh[1], Dae Seok Eom[1,2,3]*

[1]Department of Developmental and Cell Biology, University of California, Irvine, United States; [2]Center for Complex Biological Systems, University of California, Irvine, United States; [3]UC Irvine Skin Biology Resource Center, University of California, Irvine, United States

*For correspondence:
dseom@uci.edu

Competing interest: The authors declare that no competing interests exist.

## eLife Assessment

This is a **valuable** study showing that differentiated cells of the zebrafish skin form membrane protrusions called cytonemes that contact and likely transmit Notch signals to cells of the undifferentiated layer below. The data are **convincing** that cytoneme like protrusions from the periderm are required for proper periderm structure, proliferation, gene expression, and Notch signaling. Evidence that inflammatory signaling through IL-17 affects epidermal differentiation, Notch and cytoneme formation is **solid**, but whether these are through a single common or two parallel pathways requires further investigation.

**Abstract** The skin, the largest organ, functions as a primary defense mechanism. Epidermal stem cells supply undifferentiated keratinocytes that differentiate as they migrate toward the outermost skin layer. Although such a replenishment process is disrupted in various human skin diseases, its underlying mechanisms remain elusive. With high-resolution live imaging and in vivo manipulations, we revealed that Notch signaling between keratinocytes is mediated by signaling filopodia called cytonemes and is essential for proper keratinocyte differentiation and proliferation. Inhibiting keratinocyte cytonemes reduced Notch expression within undifferentiated keratinocytes, leading to abnormal differentiation and hyperproliferation, resembling human skin disease phenotypes. Over-production of Interleukin (IL)–17 signal, associated with skin diseases like psoriasis, induces psoriatic phenotypes by reducing cytoneme extension in zebrafish. Our study suggests that intercellular signaling between keratinocytes through cytonemes is critical for epidermal maintenance, and its misregulation could be an origin of human skin diseases.

## Introduction

The vertebrate skin is a multilayered structure comprised of three main layers: the epidermis, the dermis, and the hypodermis (*Akat et al., 2022*). Each layer has distinct anatomical characteristics and serves specific functions. Of these layers, the epidermis, as the outermost layer, plays a pivotal role as the body's first line of defense against environmental threats such as pathogens. It also plays a crucial role in regulating body temperature and preventing water loss into the surrounding environment. The epidermis consists of various layers, including the stratum basale, which houses a single layer of stem cells, and several layers of differentiated cells, including the stratum spinosum, stratum granulosum, and stratum lucidum. The outermost layer, the stratum corneum, primarily comprises

dead keratinocytes that are continuously shed and replenished by underlying cells differentiated from epidermal stem cells in mammals (*Segre, 2006*).

In zebrafish, the epidermis shares a similar composition to that of mammals. However, unlike mammals, aquatic animals lack a cornified layer and instead rely on a mucous layer to protect the epidermis in the aquatic environment. The basal layer of zebrafish, equivalent to the mammalian stratum basale, consists of stem cells that provide keratinocytes for replenishment during epidermal remodeling or wound healing. The periderm, comparable to the mammalian stratum granulosum and lucidum, is where fully differentiated keratinocytes are found. Similar to mammalian stratum granulosum, an intermediate layer is present between the basal layer and the periderm, housing undifferentiated keratinocytes (*Chang and Hwang, 2011*). This intermediate layer is formed during metamorphosis and persists throughout adulthood in zebrafish (*Lee et al., 2014*).

The maintenance of a healthy epidermal barrier requires the intricately controlled proliferation and differentiation of keratinocytes, which are supplied from underlying basal stem cells (*Blanpain and Fuchs, 2009*). As keratinocytes migrate apically from the stem cell layer, they undergo differentiation while concurrently exhibiting proliferation within the intermediate layer. Ultimately, fully differentiated keratinocytes replenish the periderm. This replenishment mechanism is well-conserved in both zebrafish and humans, and its misregulation constitutes a pivotal factor in the etiology of various human skin diseases, including *psoriasis*, *atopic dermatitis*, and others (*Chang and Hwang, 2011*; *Lowes et al., 2014*; *Nowell and Radtke, 2013*; *Schempp et al., 2009*; *Zhou et al., 2022*). Studies have also revealed that maintaining a healthy epidermis requires complex intercellular communications between keratinocytes and the immune system, adding complexity to our efforts to comprehensively understand the mechanism behind this process (*Blanpain and Fuchs, 2009*; *Park et al., 2021*).

One of the major signaling pathways critical for keratinocyte differentiation and proliferation is the Notch pathway. Studies have demonstrated that Notch receptors and ligands exhibit spatially restricted expression patterns in the layers of the epidermis (*Nowell and Radtke, 2013*). Notch 1, 2, and 3 are notably prevalent in the undifferentiated keratinocytes in mice (*Blanpain et al., 2006*; *Nickoloff et al., 2002*; *Thélu et al., 2002*). Moreover, dysregulated Notch signaling has been linked to abnormal keratinocyte differentiation and hyperproliferation, which are one of the hallmarks of various skin diseases including *psoriasis* and *atopic dermatitis* (*Gratton et al., 2020*), highlighting the importance of proper Notch signaling in epidermal maintenance to retain the balance between keratinocyte differentiation and proliferation (*Ota et al., 2014*; *Rangarajan et al., 2001*). Nonetheless, the precise mechanisms underlying the initiation of Notch signaling in undifferentiated keratinocytes and its dynamic regulations are not yet understood.

Furthermore, various combinations of cytokines produced by different immune cells under inflammatory conditions can significantly contribute to the onset or progression of these skin diseases. Among these, interleukin-17 (IL-17) is a well-characterized cytokine closely associated with the symptoms of these conditions. The overproduction of IL-17 by T helper (Th)17 cells and other immune cells is a primary driver of certain skin diseases, such as *psoriasis*. Biologic treatments that inhibit IL-17 have proven to effectively alleviate the symptoms of these conditions (*Mosca et al., 2021*). Unfortunately, discontinuing these treatments often leads to symptom recurrence. Also known, side effects associated with these inhibitors include headaches, nasopharyngitis, and infections (*Campa et al., 2016*). Consequently, a definitive cure remains elusive currently (*Lowes et al., 2014*; *Fragoulis et al., 2016*).

Thus, IL-17 is one of the key molecules that contribute to the development of *psoriasis* and other skin diseases. The IL-17 pathway exerts its effects on keratinocytes, endothelial cells, and immune cells, resulting in the production of additional cytokines that promote positive feedback loops, sustaining the inflammatory state (*Mosca et al., 2021*). However, the underlying cellular and molecular mechanisms of this process are not yet fully understood. Therefore, gaining a comprehensive understanding of the interplay between keratinocytes, the Notch pathway, IL-17, and other players is essential for advancing the treatment of these diseases.

Intercellular signaling plays a critical role in skin development and maintenance. In recent years, a growing body of evidence has demonstrated that cells can establish communication through long, thin cellular protrusions extended by cells involved in either sending or receiving signals. These protrusions can be categorized based on their cytoskeletal composition, mode of signal delivery, morphology, and other characteristics. Examples include cytonemes, airinemes, tunneling nanotubes, intercellular bridges, migrasomes, exophers, and more (*Eom, 2020*; *Kornberg and Roy, 2014b*; *Ma*

*et al., 2015*; *Melentijevic et al., 2017*; *Zhang and Scholpp, 2019*). These specialized cellular protrusions have been observed in diverse cell types and species, ranging from fruit flies and sea urchins to zebrafish and mice, with their signaling roles experimentally confirmed in vivo (*Eom, 2020*; *Zhang and Scholpp, 2019*; *Daly et al., 2022*; *Hall et al., 2024*; *Kornberg and Roy, 2014a*). These specialized protrusions are significantly longer than typical filopodia and establish direct contact with target cells and serve as highways for transporting signaling molecules for major signaling pathways such as Hedgehog, Wnt, TGFβ, FGF, Notch in many in vivo and in vitro contexts (*Daly et al., 2022*). These signaling cellular protrusions temporally exist and have thin actin or actin/tubulin-based filaments. Consequently, live imaging is considered one of the most effective methods for observing these structures. For that reason, zebrafish has emerged as an excellent model system for studying these cellular protrusions. The transparent nature of their early developing embryos makes zebrafish particularly well-suited for this purpose. Moreover, zebrafish are vertebrates that share common mechanisms with mammals in terms of epidermal remodeling and maintenance (*Eisenhoffer et al., 2017*; *Li et al., 2011*; *Martínez-Navarro et al., 2019*).

In this study, we show evidence that cytonemes extended by fully differentiated keratinocytes play a crucial role in activating Notch signaling in undifferentiated keratinocytes, thus contributing to the maintenance of epidermal homeostasis and remodeling in zebrafish. Furthermore, we demonstrate that IL-17, a key player in the pathogenesis of human skin diseases, and *clint1*, a gene required for epidermal homeostasis, regulate keratinocyte cytonemes (*Dodd et al., 2009*; *Sahlén et al., 2021*). Collectively, our findings shed light on the mechanisms that govern the regulation of skin replenishment through keratinocyte cytonemes, providing a novel perspective on understanding how skin diseases can originate from keratinocytes autonomously through the mediation of cellular protrusions.

## Results

### Differentiated keratinocytes extend cytoneme-like cellular protrusions

In previous studies, we have reported a unique type of signaling cellular protrusions known as 'airinemes' and have investigated their signaling roles in pigment cells of zebrafish (*Eom, 2020*; *Bowman et al., 2023*; *Eom et al., 2015*; *Eom and Parichy, 2017*; *Park et al., 2022*). To extend our understanding of airinemes in other cell types, we employed random cell labeling and identified several cell types that extend airineme-like protrusions featured by highly curved filaments and large vesicles at their tips (*Lee et al., 2014*; *Eom, 2020*; *Eom et al., 2015*). Fully differentiated keratinocytes, marked by *krt4*, were among the cell types displaying these airineme-like protrusions (*Lee et al., 2014*; *Figure 1A*, arrowhead). These keratinocytes intriguingly also extend cytoneme-like protrusions, characterized by relatively straight filaments and a lack of vesicle at the tips (*Eom, 2020*; *Kornberg and Roy, 2014b*; *Zhang and Scholpp, 2019*; *Figure 1A–B*, arrowheads). Nevertheless, we observed a low incidence of airineme-like protrusions, contrasting with the more frequent occurrence of cytoneme-like structures in these keratinocytes (*Figure 1C*). Consequently, we conducted a more in-depth characterization of these cytoneme-like protrusions. It is noted that cytonemes primarily consist of actin filaments, whereas airinemes are both actin and tubulin-based structures (*Eom, 2020*). To affirm their nature, we utilized an actin marker, *LifeAct-mRuby*, which clearly labeled these cytoneme-like protrusions while showing no co-localization with *tubulin-mCherry* (*Figure 1D–E*). Additionally, cytoneme-like protrusions were significantly reduced upon treatment with a cdc42 inhibitor, which inhibits actin polymerization (*Figure 1—figure supplement 1*). Therefore, based on their cytoskeletal composition and morphology, we classified them as cytonemes.

We observed that keratinocytes marked by *krt4* expression extend cytonemes with the highest frequency during post-embryonic stages, specifically between 6.5–7.5 SSL (Standardized Standard Length; *Parichy et al., 2009*; *Figure 1C*). It is worth noting that embryonic keratinocytes are replaced by post-embryonic cells around these stages, suggesting a potential role of cytonemes in epidermal remodeling and maintenance (*Lee et al., 2014*; *Parichy et al., 2009*). These keratinocyte cytonemes exhibit an approximate speed of 2.98 µm/min during extension and retract at 1.9 µm/min, with an average length of 18.21 µm (*Figure 1F–G*).



**Figure 1.** Differentiated keratinocytes extend cytoneme-like cellular protrusions. (**A**) Airineme-like protrusion (arrowhead), and (**B**) cytoneme-like protrusions (arrowheads) labeled in *krt4:palmEGFP* injected keratinocytes. (**C**) Keratinocytes expressing *krt4* extend cytoneme-like protrusions most frequently during metamorphic stages (6.5–7.5 SSL; N=457 cells, 19 larvae total). A cytoneme-like protrusion (arrowhead) labeled for (**D**) F-actin, revealed by *LifeAct-mRuby* but did not colocalize (**E**) with tubulin, revealed by *Tuba-mCherry*. (**F**) Keratinocyte cytonemes extend faster than they retract (N=125 cells, 4 larvae total), with an average length (**G**) of 18.21 µm (N=190 cells, 4 larvae total). Scale bars: 20 µm (**A**, **B**, **D**, **E**). Error bars indicate mean ± SEM.

The online version of this article includes the following figure supplement(s) for figure 1:

**Figure supplement 1.** Effect of actin inhibitor treatment on cytoneme extension.

## Keratinocyte cytonemes establish physical contact with underlying undifferentiated keratinocytes

Similar to mammals, the zebrafish epidermis consists of three layers (*Chang and Hwang, 2011*). We specifically visualized the periderm, which is the outermost layer and consists of *krt4+* fully differentiated keratinocytes. The intermediate layer, where undifferentiated keratinocytes reside, was marked with *krtt1c19e* promoter driving tdTomato expression, pseudo-colored in magenta (*Chang and Hwang, 2011*; *Lee et al., 2014*). The basal layer was also labeled with *krtt1c19e* but exhibited a typical polygonal morphology (*Figure 2A–A"*). This allowed us to distinguish all three epidermal layers using the *Tg(krt4:lyn-EGFP; krtt1c19e:lyn-tdTomato)*.

In addition to *krt4+* keratinocytes, we explored whether undifferentiated keratinocytes (*krtt1c19e+*) also extend cytonemes or other types of cellular protrusions during metamorphic stages. However, we observed that cytoneme-like protrusions from undifferentiated keratinocytes were significantly fewer in numbers compared to those from fully differentiated keratinocytes, and we did not observe cytoneme-like protrusions from basal stem cells (*Figure 2B–C*). In this study, our focus was primarily on cytonemes from *krt4+* differentiated keratinocytes.

To investigate the role of keratinocyte cytonemes, we first asked what the target cells of those cytonemes are. High-resolution confocal imaging revealed that cytonemes originating from peridermal keratinocytes (*krt4+*) establish physical contact with underlying undifferentiated keratinocytes (*krtt1c19e+*) in the intermediate layer. Cross-sectional views showed that the tips of these cytonemes stop at the intermediate layer, with no observed instances reaching the basal layer (*Figure 2D–D"'*, red dashed circles).

## Cytoneme-mediated signaling regulates keratinocyte differentiation and proliferation

The observations that cytonemes are most often observed during epidermal remodeling and their target cells are undifferentiated keratinocytes led us to hypothesize that cytoneme-mediated signaling plays a pivotal role in regulating keratinocyte differentiation and proliferation.

To test this hypothesis, we generated a transgenic line capable of temporally expressing a dominant negative form of cdc42 (cdc42DN) in peridermal keratinocytes, a strategy frequently used to inhibit cytoneme extension in various contexts (*Zhang and Scholpp, 2019*; *Daly et al., 2022*). We implemented a dually inducible TetON system in the transgene, denoted as *Tg(krt4:TetGBDTRE-v2a-cdc42DN)*, and immersed the transgenic and control fish in water containing inducer drugs, doxycycline and dexamethasone, for temporal cdc42DN transgene induction (*Eom et al., 2015*; *Knopf et al., 2010*). To minimize potential off-target effects of cdc42 manipulation, we determined the drug concentrations at which normal filopodial or lamellipodial activities were maintained while inhibiting cytoneme extensions (*Video 1*). We employed two control groups in this experiment: one consisting of a transgenic line with no drugs and the other comprising non-transgenic fish treated with the same drug concentrations as the experimental group. Under these conditions, we observed a significant reduction in the frequency of cytoneme extension in the cdc42DN-expressing transgenic line within the drug-treated group (*Figure 3A–A" and B*). In contrast to the two control groups, the peridermal layer in the cytoneme-inhibited group displayed severe disorganization (*Figure 3C–C"*). Furthermore, the expression of the undifferentiated keratinocyte marker, *krtt1c19e* (magenta), was dramatically increased in the periderm when cytoneme extension was inhibited (*Figure 3C–C" and D*). Although the cytoneme inhibition is evident after overnight treatment with the inducing drugs, noticeable epidermal phenotypes begin to appear after 3 days of treatment. This reflects the higher cytoneme extension frequency and their potential role during metamorphic stages, which takes a couple of weeks (*Figure 1C*; *Lee et al., 2014*; *Parichy et al., 2009*).

This co-expression of differentiated and undifferentiated keratinocyte markers in peridermal keratinocytes, along with the data that the intermediate layer is not depleted in cytoneme-inhibited animals, suggests that cytoneme-mediated signal is critical for the terminal differentiation of keratinocytes (*Figure 3C–D*).

We mosaically expressed *cdc42DN* in peridermal keratinocytes and observed localized disorganization in the periderm, along with co-expression of *krt4* and *krtt1c19e* in peridermal keratinocytes (*Figure 3—figure supplement 1A, D, E-E'*). This suggests that these defects result from the loss of cytonemes rather than broad *cdc42* inhibition. We also locally expressed other dominant-negative



**Figure 2.** Keratinocyte cytonemes establish physical contact with underlying undifferentiated keratinocytes. (**A**–**A″**) Postembryonic zebrafish epidermis of *Tg(krt4:lyn-EGFP;krtt1c19e:lyn-tdTomato)* comprises three distinct layers. (**A**) Expression of *lyn-EGFP* in *krt4+* cells in the periderm layer. (**A′**–**A″**) Expression of *lyn-tdTomato* (pseudo-colored magenta) in *krtt1c19e+* cells in (**A′**) the intermediate layer and (**A″**) the basal layer. Note that the green circular labeling in (**A′**, arrowheads) represents mucous-secreting goblet cells. *Chang and Hwang, 2011* (**B**, **C**) Undifferentiated keratinocytes (KC) extend significantly fewer cytoneme-like protrusions (arrowheads, p<0.0001, N=49, 3 larvae; N=91 cells, 6 larvae total). (**D**–**D″′**) Cytonemes from fully differentiated keratinocytes make physical contact with undifferentiated keratinocytes in the intermediate layer (red dotted circles). Note that dotted lines indicate the layers in the cross-sectional views. Statistical significances were assessed by Student t test. Scale bars: 20 µm (**A**–**A″**, **C**, **D**–**D″′**). Error bars indicate mean ± SEM.



**Video 1.** Cytoneme extension in keratinocytes. In cdc42DN-expressing *krt4+* (green) keratinocytes (right), cytoneme extension is significantly inhibited. It is worth noting that normal filopodial and lamellipodial extension seems unaffected. In the control groups (left and middle), cytonemes actively extend. The movie was captured at 3 min intervals, and the hours and minutes elapsed are displayed in the upper right corner.
https://elifesciences.org/articles/97400/figures#video1

forms of small GTPases, such as Rac1 or Rhoab. Interestingly, *krt4+* keratinocytes expressing dominant negative *rac1* exhibited a significant reduction in cytoneme extension and epidermal defects similar to those seen with *cdc42* inhibition, while *rhoab* manipulation showed no effects. This indicates that the observed defects are highly likely related to cytoneme loss rather than specifically to *cdc42* inhibition (***Figure 3— figure supplement 1B, D, F-F'***; ***Figure 3—figure supplement 1C, D, G-G'***).

Additionally, we counted the number of peridermal keratinocytes within a 290µm x 200µm rectangle and found a substantial increase in the cytoneme-inhibited group (***Figure 3E***). We also confirmed that the number of Edu+ cells is significantly increased in this group (***Figure 3F–G***). However, it is important to note that cytoneme inhibition did not affect the cell death rate or the number of goblet cells (***Figure 3H***, ***Figure 3—figure supplement 2***). These findings strongly suggest that cytoneme-mediated signaling controls keratinocyte differentiation and proliferation.

## Cytonemes activate Notch in undifferentiated keratinocytes

Previous studies have established the role of Notch signaling in controlling keratinocyte differentiation and proliferation. However, the mechanisms triggering Notch signaling in keratinocytes have remained obscure (***Rangarajan et al., 2001***; ***Moriyama et al., 2008***; ***Nguyen et al., 2006***). To investigate whether the target cells of cytonemes, specifically undifferentiated keratinocytes in the intermediate layer, are responsive to Notch signaling, we crossed Notch signal reporter line, *Tg(Tp1:H2B-mCherry)*, with *Tg(krt4:lyn-EGFP; krtt1c19e:lyn-tdTomato)* to label all three epidermal layers. In this Notch reporter line, the promoter from the Epstein Barr Virus terminal protein 1 (*Tp1*) gene was used as a Notch responsive element, containing two Rbp-Jκ binding sites that drive expression of Histone 2B-Cherry (***Parsons et al., 2009***). We observed that only undifferentiated keratinocytes in the intermediate layer exhibit Notch responsiveness, whereas keratinocytes in the periderm or basal layer do not (***Figure 4A***, arrowheads, ***Figure 4—figure supplement 1***). This finding was further confirmed with immunostaining against Her6, a Notch effector, which revealed its specific expression in the intermediate layer (***Figure 4B***; ***Liu et al., 2006***).

Subsequently, we tested whether Notch activation in undifferentiated keratinocytes is dependent on cytonemes. First, we validated the *Tp1* Notch reporter line, *Tg(Tp1:EGFP)*, by measuring cytoplasmic EGFP signal before and after Notch inhibitor LY411575 treatment, which demonstrated a significant decrease in Notch signaling, as expected (***Figure 4—figure supplement 2A, B***). Next, we measured the difference in *Tp1* intensity before and after cytoneme inhibition. Notably, we observed a significant reduction in Notch signal under conditions where cytonemes were compromised (***Figure 4C***, ***Figure 4—figure supplement 2C***). Similar to the effects on the epidermis after cytoneme inhibition (***Figure 3***), it takes 3 days to observe a significant reduction in Notch signal in the undifferentiated keratinocytes. We also found more abundant Notch-activated keratinocytes during metamorphic stages (***Figure 4—figure supplement 2D***, ***Figure 1C***).

Next, we predicted that if cytonemes contribute to transmitting a Notch-Delta signal from differentiated keratinocytes to undifferentiated keratinocytes, then cytonemes should contain Delta ligands. Upon detecting DeltaC mRNA expression in the cytoneme extending keratinocytes but not in its target cells (***Figure 4—figure supplement 1***), we investigated the localization of DeltaC protein within the cytonemes of the *krt4+* differentiated keratinocytes. For this experiment, we utilized a construct where a 109 kb BAC containing zebrafish *dlc* coding sequence regulatory elements that were recombineered to generate a mCherry fusion, resulting in *dlc:Dlc-mCherry* (***Eom et al., 2015***). We confirmed the presence of Dlc-mCherry along the cytonemes (***Figure 4D***, yellow arrowheads). These findings suggest that cytonemes are responsible for Notch activation in undifferentiated keratinocytes.



**Figure 3.** Cytoneme-mediated signaling regulates keratinocyte differentiation and proliferation. (**A–A''**) Still images from time-lapse movies demonstrate that cytoneme extension (arrowheads) is inhibited in keratinocytes expressing cdc42DN. (**B**) Expression of *cdc42DN* effectively inhibits cytoneme extension ($F_{2, 231}$ = 25.38, p<0.0001, N=234 cells, 10 larvae total). (**C**) Periderm layer of *Tg(krt4:TetGBDTRE-v2a-cdc42DN;krt4:lyn-EGFP;krtt1c19e:lyn-tdTomato)* without drug treatment. (**C'**) Periderm layer of *Tg(krt4:lyn-EGFP;krtt1c19e:lyn-tdTomato)* with drug treatment.

*Figure 3 continued on next page*

*Figure 3 continued*

(**C″**) Periderm layer of *Tg(krt4:TetGBDTRE-v2a-cdc42DN;krt4:lyn-EGFP;krtt1c19e:lyn-tdTomato)* with drug treatment exhibits disorganization. Intermediate keratinocytes were not depleted, and basal stem cells were unaffected by the manipulation. Note that, unlike the other two controls, the epidermis was not flat, resulting in partial display of basal cells in the intermediate layer and dark areas in the basal layer. Cross-section view exhibits *krtt1c19e* expression at the periderm. (**D**) Cytoneme inhibition increases *krtt1c19e* expression in the periderm ($F_{2, 387}$ = 226.7, p<0.0001, N=39 larvae total) and leads to an increased number of keratinocytes within the periderm (**E**) ($F_{2, 18}$ = 27.36, p<0.0001, N=21 larvae total). (**F–G**) Significant increase in proliferating peridermal keratinocytes in cdc42DN expressing animals ($F_{2, 28}$ = 4.888, p=0.0151, N=9 larvae). (**H**) The cell death rate in the periderm is not affected by cytoneme inhibition, tested with acridine orange incorporation ($F_{2, 17}$ = 3.192, p=0.0666, N=20 larvae total). Statistical significances were assessed by One-way ANOVA followed by Tukey's HSD post hoc test. Scale bars: 20 μm (**A–A″**, **C–C″**, **F**). Error bars indicate mean ± SEM.

The online version of this article includes the following figure supplement(s) for figure 3:

**Figure supplement 1.** Local manipulations with dominant-negative cdc42, rac1, or rhoab in peridermal keratinocytes.

**Figure supplement 2.** Goblet cell counts in the zebrafish epidermis.

We then asked why Notch activation is dependent on cytonemes since peridermal keratinocytes and underlying intermediate keratinocytes are in constant contact with each other. We revealed that intermediate keratinocytes exclusively express *lunatic fringe* (*lfng*), which potentially act as a default inhibitor of Notch activation in these cells (*Dale et al., 2003*; *Panin et al., 1997*; *Figure 4—figure supplement 3*). However, this needs further studies to understand how cytonemes overcome this Notch signaling barrier or potentiate the signal.

Next, we hypothesized that if Notch activation by cytonemes is essential for keratinocyte differentiation and proliferation, then direct inhibition of Notch signal in undifferentiated keratinocytes should yield phenotypes similar to those observed when cytonemes are inhibited (*Gratton et al., 2020*). To investigate this, we generated a transgenic line capable of temporally expressing dominant negative form of Suppressor of Hairless (SuHDN) in the undifferentiated keratinocytes, *Tg(krtt1c19e:TetGBDTRE-v2a-SuHDN)* (*Eom et al., 2015*). We observed that Notch inhibition in the undifferentiated keratinocytes resulted in a disorganized periderm, abnormally increased expression of undifferentiated keratinocyte marker, and hyperproliferation of keratinocytes in the periderm (*Figure 4E-E″, F-H*, *Figure 4—figure supplement 2C*). These results closely mirrored the phenotypes observed when we inhibited cytoneme extension in peridermal keratinocytes (*Figure 3C-C″, D-E*, *Figure 4—figure supplement 2C*). These observations suggest that Notch signaling, activated by cytonemes in undifferentiated keratinocytes, plays a critical role in epidermal remodeling and maintenance.

## Interleukin-17 regulates keratinocyte cytonemes

The aberrant differentiation and hyperproliferation of keratinocytes observed when cytoneme extension was inhibited in peridermal keratinocytes (*krt4+*) or Notch signaling was inhibited in undifferentiated keratinocytes (*krtt1c19e+*) are hallmark features of many human skin diseases such as *psoriasis*, *atopic dermatitis,* and more (*Gratton et al., 2020*; *Armstrong and Read, 2020*; *Figures 3 and 4*). Also, it has been well established that impaired Notch signaling is associated with some of these human skin disorders (*Nowell and Radtke, 2013*; *Gratton et al., 2020*). Thus, we asked which key molecules are critical for the onset or progression of these skin diseases and whether they have an impact on keratinocyte cytoneme signaling.

One of them is Interleukin-17 (IL-17), and we hypothesized IL-17 signaling could influence cytoneme extension in peridermal keratinocytes. Given that IL-17 is a cytokine released into the extracellular space, and to test its role in peridermal keratinocytes, we decided to overexpress the IL-17 receptors. It is noted that Interleukin-17 receptors and ligands are evolutionarily conserved across chordates (*González-Fernández et al., 2020*; *Wu et al., 2011*).

To test, we generated transgenic lines that cell-autonomously overexpressed il17 receptor D (*il17rd*) or il17 receptor A1a (*il17ra1a*) in fully differentiated peridermal keratinocytes (*krt4+*), *Tg(krt4:nVenus-v2a-il17rd)* and *Tg(krt4:il17ra1a-v2a-mCherry)*. Both of these IL-17 receptor-overexpressing transgenic lines exhibited a significant reduction in cytoneme extension, resulting in a decreased Notch activation in undifferentiated keratinocytes (*Figure 5A–C*, *Figure 4—figure supplement 2E*, *Figure 5—figure supplement 1A–B*). Moreover, similar to the two previous manipulations, we observed a severely disorganized periderm, an upregulation of the undifferentiated keratinocyte marker expression, and hyperproliferation of keratinocytes within the periderm (*Figure 5D–D″ and E–H*).



**Figure 4.** Cytonemes activate Notch in undifferentiated keratinocytes. (**A**) Postembryonic zebrafish epidermis of *Tg(krt4:lyn-EGFP;krtt1c19e:lyn-tdTomato;TP1:H2BmCherry)* shows Notch responsiveness in undifferentiated keratinocytes within the intermediate layer but not in the periderm or basal layer. (**B**) Zebrafish Her6 protein expression in *krtt1c19e+*intermediate keratinocytes. A cross-sectional view of *Tg(krtt1c19e:tdtomato)* epidermis

*Figure 4 continued on next page*

*Figure 4 continued*

shows that Her6 expression overlaps with the intermediate layer marker. (**C**) Inhibition of cytonemes reduces Notch responsiveness in undifferentiated keratinocytes ($F_{2, 33}$ = 48.27, p < 0.0001, N=36 larvae total) (Figure S5C). (**D**) Expression of DeltaC-mCherry fusion protein along the cytoneme (yellow arrowheads). (**E–E″**) Notch inhibition by expressing SuHDN in undifferentiated keratinocytes results in the disorganization of periderm (**E″**). (**E**) Properly organized periderm layer of *Tg(krtt1c19e:tetGBDTRE-v2a-SuHDN;krt4:lyn-EGFP;krtt1c19e:lyn-tdTomato)* without drug treatment and (**E′**) *Tg(krt4:lyn-EGFP;krtt1c19e:lyn-tdTomato)* with drug treatment. (**F**) Notch inhibition increases *krtt1c19e* signal in the periderm ($F_{2, 397}$ = 89.47, p < 0.0001, N=40 larvae total), (**G**) reduces Notch responsiveness in undifferentiated keratinocytes ($F_{2, 32}$ = 13.18, p < 0.0001, N=35 larvae total), and leads to (**H**) an increased number of keratinocytes in the periderm ($F_{2, 19}$ = 6.620, p = 0.0066, N=22 larvae total). Statistical significances were assessed by One-way ANOVA followed by Tukey's HSD post hoc test. Scale bars, 20μm (**A**, **C**, **D**). Error bars indicate mean ± SEM.

The online version of this article includes the following source data and figure supplement(s) for figure 4:

**Figure supplement 1.** Gene expressions in keratinocytes.

**Figure supplement 1—source data 1.** Original gel images for RT-PCR analysis displayed in *Figure 4—figure supplement 1*, with labels.

**Figure supplement 1—source data 2.** Original files for RT-PCR analysis displayed in *Figure 4—figure supplement 1*, without labels.

**Figure supplement 2.** Notch reporter (*Tp1*) expression.

**Figure supplement 3.** Notch signal modifiers in undifferentiated keratinocytes.

---

To assess whether IL-17 signaling regulates keratinocyte differentiation and proliferation through cytonemes, first we overexpressed the *il17a* ligand in the epidermis and examined its effect on cytoneme extension. Given the abundance of tissue-resident macrophages in the epidermis, we generated a transgenic line that overexpressed *il17a* under a macrophage promoter, *Tg(mpeg1:il17a-v2a-mCherry)* (*Figure 5—figure supplement 1C*; *Bowman et al., 2023*). Consistent with the experiments that overexpressed il17 receptors, we confirmed that *il17a* ligand overexpression in the epidermal microenvironment led to a significant reduction in cytoneme extension in the differentiated keratinocytes (*Figure 6A–B*). Next, we generated an *il17a* loss-of-function mutant using the CRISPR/Cas9 approach. This mutant harbors an 11 bp deletion in the second exon of the *il17a* gene, resulting in a premature stop codon (*Figure 6—figure supplement 1*). We observed a significant increase in cytoneme extension frequency in *il17a* heterozygote or homozygote mutants as compared to their wild-type siblings (*Figure 6C–D*). Together, our findings suggest that IL-17 signaling controls cytoneme extension.

Subsequently, we sought to understand how the IL-17 signal impacts cytoneme extension. Studies have shown that IL-17 signaling can regulate actin cytoskeleton in keratinocytes (*Borow-czyk et al., 2020*; *Das et al., 2019*). To evaluate whether *il17rd* or *il17ra1a* overexpression alters the actin cytoskeleton and consequently cytonemes, we compared keratinocyte microridges, which are actin cytoskeleton-rich structures, between control and *il17rd* or *il17ra1a* overexpressed peridermal keratinocytes (*Pinto et al., 2019*; *van Loon et al., 2020*). We first confirmed that overexpression of *cdc42DN* or *rac1DN* disrupts microridge formation in differentiated keratinocytes (*Figure 6—figure supplement 2*). Our observation revealed severe disorganization of microridges in the keratinocytes overexpressing *il17rd or il17ra1a*. This was coupled with a significant down-regulation of *Cdc42* and *Rac1* GTPase expression in these cells (*Figure 6E–G*). These data suggest that IL-17 signaling negatively regulates cytoneme extension by down-regulating *Cdc42* and *Rac1*, which are essential for controlling the actin cytoskeleton (*Figure 3—figure supplement 1*, *Figure 6—figure supplement 2*; *Daly et al., 2022*; *Hall, 1998*; *Jaffe and Hall, 2005*; *Stanganello et al., 2015*).

We then crossed two transgenic lines: *Tg(krt4:TetGBDTRE-v2a-cdc42DN)* and either *Tg(krt4:nVenus-v2a-il17rd)* or *Tg(krt4:il17ra1a-v2a-mCherry)*. Given that il17 receptor overexpression reduced *Cdc42* expression in differentiated keratinocytes (*Figure 6E*), we anticipated observing a more severe inhibition of cytoneme extension when these manipulations are combined compared to individual manipulations. Indeed, when inhibiting cytoneme extension with the *cdc42DN* along with *il17rd* or *il17ra1a* overexpression, we observed a more significant inhibition of cytoneme extension than when manipulated individually (*Figure 6H*).



**Figure 5.** Disruption of epidermal maintenance by Interleukin-17 receptor overexpression. Overexpression of *il17rd* or *il17ra1a* in *krt4+* keratinocytes results in (**A**, **B**) a significant reduction in cytoneme extension ($F_{2,325}$ = 38.48, p < 0.0001, N=328 cells, 10 larvae total), and (**C**) decreased Notch responsiveness ($F_{2,34}$ = 23.16, p < 0.0001, N=37 larvae total) (Figure S5E). (**D–D"**) The periderm is disrupted in (**D'**) *Tg(krt4:nVenus-v2a-il17rd;krt4:lyn-EGFP;krtt1c19e:lyn-tdTomato)* and (**D"**) *Tg(krt4:nVenus-v2a-il17ra1a;krt4:lyn-EGFP;krtt1c19e:lyn-tdTomato)* compared to the (**D**) properly arranged

*Figure 5 continued on next page*

*Figure 5 continued*

periderm of *Tg(krt4:lyn-EGFP;krtt1c19e:lyn-tdTomato)*, resulting in (**D'**, cross-section) an increased *krtt1c19e* signal in the periderm (**E**) ($F_{2, 287}$ = 109.7, p < 0.0001, N=29 larvae total) and (**F**) an increased number of keratinocytes in the periderm ($F_{2, 15}$ = 27.79, p < 0.0001, N=18 larvae total). (**G**, **H**) Edu+ peridermal keratinocytes are significantly increased in il17 receptor overexpressed larvae ($F_{2, 11}$=8.259, p=0.0065, N=14 larvae total). Statistical significances were assessed by One-way ANOVA followed by Tukey's HSD post hoc test. Scale bars: 20μm (**A**, **D**, **G**). Error bars indicate mean ± SEM.

The online version of this article includes the following figure supplement(s) for figure 5:

**Figure supplement 1.** Overexpression of *il17rd, il17ra1a,* or *il17a* in transgenic animals.

Our data collectively support the idea that cytoneme-mediated intercellular signaling between peridermal and underlying undifferentiated keratinocytes is essential for epidermal remodeling and maintenance. IL-17 functions as a regulator capable of affecting cytoneme extension by modulating the actin cytoskeleton in keratinocytes. Consequently, our data suggest that the overproduction of IL-17 ligands by immune cells in conditions such as psoriatic or other skin diseases leads to epidermal imbalance through altered cytoneme signaling in keratinocytes.

In addition, recent human Genome-Wide Association Studies (GWAS) have identified a close association between *clint1* and individuals with *psoriasis* and *atopic dermatitis* (*Sahlén et al., 2021*). The clathrin interactor 1 (*clint1*), also referred to as enthoprotin and epsinR, functions as an adaptor molecule that binds SNARE proteins and plays a role in clathrin-mediated vasicular transport (*Wasiak et al., 2002*). It has also been reported that *clint1* is expressed in epidermis and plays an important role in epidermal homeostasis and development in zebrafish (*Dodd et al., 2009*). Given these reported findings, we investigated to determine whether *clint1* plays a role in epidermal maintenance through its influence on keratinocyte cytonemes. Indeed, we found that *clint1* mutants exhibited a significant decrease in cytoneme extension in peridermal keratinocytes, accompanied by reduced Notch signaling in undifferentiated keratinocytes. As a result, these mutant fish displayed abnormal keratinocyte differentiation and hyperproliferation of keratinocytes (*Figure 6—figure supplement 3*). These findings suggest that *clint1* is yet another regulator of cytoneme-mediated intercellular signaling in keratinocytes.

In conclusion, the intercellular signaling facilitated by cytonemes between fully differentiated and undifferentiated keratinocytes is crucial for the maintenance and remodeling of the epidermis. Our study implies that the patients with human skin diseases may have defective cytoneme signaling between keratinocytes.

## Discussion

It is intriguing to observe that keratinocytes have the ability to extend both cytoneme and airineme-like protrusions. While these protrusions differ in their extension frequency and developmental context, this observation implies that a single cell type can employ two distinct types of cellular protrusions for signaling, each capable of transferring unique signaling molecules to potentially different targets. Previous research in *Drosophila* has demonstrated that the air sac primordium (ASP) can extend multiple cytonemes that appear morphologically identical but convey different signaling molecules (*Roy et al., 2011*). However, it remains unknown whether the same cell type can extend two morphologically (and functionally) distinct protrusions that might also serve distinct functions in separate contexts. Consequently, future studies aimed at unraveling the regulation of these two types of protrusions could provide insights into how cells choose their signaling modalities based on their specific signaling requirements and environmental cues.

We have asked why cytoneme-mediated signaling is necessary between fully differentiated and undifferentiated keratinocytes, even though they are separated by opposing cell membranes. Notably, we have identified the Notch signaling pathway as being involved, which operates through membrane receptor and ligand interactions between adjacent cells. This led us to hypothesize that Notch signaling between cells in these layers is suppressed by Fringe proteins, which modulate Notch signaling. In our research, we have discovered that *lunatic fringe (lfng)* is exclusively expressed in the intermediate layer in zebrafish (*Panin et al., 1997*, *Figure 4—figure supplement 3*). Early studies in mice have also reported that three fringe genes exhibit differential expression in different epidermal layers (*Thélu et al., 1998*). Hence, it appears that Notch signaling is tightly regulated or inhibited by default between epidermal layers in both zebrafish and mice. While it is well documented that Notch



**Figure 6.** Interleukin-17 regulates keratinocyte cytonemes. (**A**) Representative images of *krt4:palmEGFP* injected cells in control or *Tg(mpeg1:il17a)*. (**B**) Significant reduction in cytoneme extension in response to overexpressed *il17a* ligands in the epidermis (p<0.0001, N=3 larvae per group). (**C**) Representative images of *krt4:palmEGFP* injected keratinocytes in WT or *il17a-/-* mutants. (**D**) Cytoneme extension frequency increased significantly in *il17a*$^{+/-}$ or *il17a*$^{-/-}$ mutants as compared to their wild-type siblings ($F_{2, 247}$=17.85, p<0.0001, N=250 cells, 9 larvae total). (**E**) FACS-sorted *krt4+* peridermal cells from *Tg(krt4:palmEGFP; krt4:il17rd)* or *Tg(krt4:palmEGFP; krt4:il17ra1a)* show significantly reduced *Cdc42* (p=0.0341 for *il17rd*, p=0.01 for *il17ra1a*, N=3 independent experiments) and Rac1 expression (p=0.0189 for *il17rd*, p=0.0017 for *il17ra1a*, N=4 independent experiments). (**F**) *il17rd-*

*Figure 6 continued on next page*

*Figure 6 continued*

or *il17ra1a* overexpressing *krt4+* keratinocytes exhibit disorganization of microridges (arrowheads), with (**G**) significantly reduced branch lengths of microridges in the transgenic animals ($F_{2,2311}$=11.11, p<0.0001, N=68 cells, 9 larvae total). (**H**) Simultaneous overexpression of *cdc42DN* and *il17rd* or *il17ra1a* in *krt4+* keratinocytes further reduces cytoneme extension (*il17rd:* p<0.0001, N=164 cells, 7 larvae total, *il17ra1a:* p<0.0001, N=115 cells, 6 larvae total). Statistical significances were assessed by One-way ANOVA followed by Tukey's HSD post hoc test or Student t test. Scale bars: 20 µm (**A**, **C**, **F**). Error bars indicate mean ± SEM.

The online version of this article includes the following source data and figure supplement(s) for figure 6:

**Figure supplement 1.** CRISPR/Cas9 induced knockout of zebrafish *il17a* resulting in a premature stop codon.

**Figure supplement 1—source data 1.** Original blots for western analysis displayed in *Figure 6—figure supplement 1*, with labels.

**Figure supplement 1—source data 2.** Original files for western analysis displayed in *Figure 6—figure supplement 1*, without labels.

**Figure supplement 2.** Microridge disorganization in *cdc42DN* or *rac1DN* overexpressed keratinocytes.

**Figure supplement 3.** *clint1* regulates epidermal maintenance via keratinocyte cytonemes.

signaling is essential for keratinocyte differentiation and proliferation, the mechanism by which Notch signaling is initiated remains unclear. Our study suggests that cytonemes from fully differentiated keratinocytes in the periderm can activate Notch signaling in undifferentiated keratinocytes in the intermediate layer. However, further investigations are necessary to fully comprehend how cytoneme-mediated signaling coordinates with Lunatic fringe or other Fringes to trigger the Notch signal.

It is widely believed that the overproduction of IL-17 is a major contributing factor to *psoriasis*, and many studies have demonstrated that IL-17 signaling triggers inflammation (*Liu et al., 2020*). IL-17 affects various cellular targets, such as keratinocytes, neutrophils, endothelial cells, promoting tissue inflammation (*Blauvelt and Chiricozzi, 2018*). Thus, further studies are essential to determine whether uncontrolled cytoneme-mediated signaling in keratinocytes correlates with the onset or progression of psoriatic or other skin diseases. Nevertheless, our data strongly indicate that cell-autonomous overexpression of *il17rd* in peridermal keratinocytes (*krt4+*) can induce psoriasis-like phenotypes by inhibiting cytoneme extensions, which subsequently leads to a reduction in Notch signaling in undifferentiated keratinocytes. A diminished Notch signal has been observed in psoriatic lesions (*Ota et al., 2014*). Our study suggests that cytoneme-mediated intercellular communication between keratinocytes plays a crucial role in epidermal maintenance, and its dysregulation may contribute to the development of human skin diseases. This idea has yet to be fully appreciated, and it could open up new avenues for understanding the onset or progression of human skin diseases.

Moreover, we have demonstrated that IL-17 can influence cytoneme extension by regulating Cdc42 GTPases, ultimately affecting actin polymerization. Consequently, it would be intriguing to investigate whether genes associated with *psoriasis* or other human skin diseases also function as regulators of cytonemes. For instance, the epidermal-specific deletion of inhibitor of NF-kB kinase 2 (IKK2) or the double knockout of the c-Jun and JunB genes in mice resulted in psoriasis-like phenotypes in a T-cell-independent manner (*Pasparakis et al., 2002*; *Zenz et al., 2005*). Among the various roles of IKK and JunB, they are also known to regulate the actin cytoskeleton (*Dubin-Bar et al., 2008*; *Lennikov et al., 2018*; *Otani et al., 2016*). Thus, it is conceivable that dysregulated cytoneme-mediated signaling may contribute to psoriatic conditions in these contexts.

Similarly, we have shown that *clint1* mutants exhibited compromised cytoneme signaling and keratinocyte hyperproliferation (*Figure 6—figure supplement 3*; *Dodd et al., 2009*). Clint1 plays an important role in vesicle trafficking, and it is suggested that endocytic pathways are critical for multiple steps in cytoneme-mediated morphogen delivery (*Daly et al., 2022*; *Kalthoff et al., 2002*). Therefore, it is interesting to investigate whether Clint1 functions as a cytoneme regulator could provide valuable insights.

Another important question requires an answer: the proper remodeling and maintenance of the epidermis seemingly depend on a well-controlled supply of keratinocytes from basal stem cells. Our study demonstrated the relay of cytoneme-mediated signals from the periderm to the underlying undifferentiated keratinocytes, leading to their differentiation and subsequent replacement of peridermal keratinocytes. However, the mechanism through which basal stem cells detect the need for an increased population of keratinocytes to maintain the intermediate layer remains unclear. This process does not necessarily involve cellular protrusion-mediated signaling. However, we discovered cytoneme-like protrusions originating from undifferentiated keratinocytes (*Figure 2B–C*). Our

preliminary data, not presented in this manuscript, suggests that these cellular protrusions project downward, indicating their interaction with basal stem cells. Although further investigation is needed, it is conceivable that a replenishment signal from external sources or the periderm can be transmitted to stem cells through a multi-step cytoneme-mediated intercellular signaling pathways.

## Experimental model and subject details

### Zebrafish

Adult zebrafish were maintained at a constant temperature of 28.5°C under a 16:8 (L:D) cycle. The zebrafish used in this study were wild-type AB$^{wp}$ or its derivative WT(ABb). Additionally, transgenic lines used include *Tg(krt4:lyn-EGFP)$^{sq18}$*, *Tg(krtt1c19e:lyn-tdTomato)$^{sq16}$* provided by T. Carney; *Tg(Tp1:EGFP)$^{um13}$*, *Tg(Tp1:H2BmCherry)$^{jh11}$* provided by M. Parsons; *clint1a$^{hi1520Tg/+}$(AB)*, obtained from the Zebrafish International Resource Center (ZIRC). The sexes of individual fish could not be distinguished since experiments were performed before developing secondary sexual characteristics. All zebrafish stocks used in this study were known to produce balanced sex ratios, ensuring that experiments sampled similar numbers of males and females. All animal work conducted in this study received approval from the University of California Irvine Institutional Animal Care and Use Committee (protocol #AUP-25–002) and adheres to institutional and federal guidelines for the ethical use of animals.

## Methods

**Key resources table**

| Reagent type (species) or resource | Designation | Source or reference | Identifiers | Additional information |
|---|---|---|---|---|
| Gene (*Danio rerio*) | il17rd | ZFIN | ZDB-GENE-020320–5 | Amplified from cDNA |
| Gene (*Danio rerio*) | il17ra1a | ZFIN | ZDB-GENE- 070705–242 | Amplified from cDNA |
| Gene (*Danio rerio*) | il17a/f1 | ZFIN; NCBI | GenBank accession number NM_001020787; ZDB-GENE-061031–3 | Amplified from cDNA |
| Antibody | anti-β-actin (rabbit monoclonal) | GeneTex | GTX637675; RRID:AB_3073746 | 1:5000 |
| Antibody | anti-IL-17 a/f1 (rabbit polyclonal) | KINGFISHER BIOTECH, INC | KP1239Z-100 | 1:1000 |
| Antibody | anti-Hes1 (rabbit monoclonal) | Invitrogen | MA5-32258; RRID:AB_2809544 | 1:250 |
| Antibody | Donkey anti-rabbit IgG H&L (Alexa Fluor 488) | Abcam | ab150073; RRID:AB_2636877 | 1:250 |
| Strain, strain background (*Danio rerio*) | WT(ABb) | PMID:26701906 | RRID:ZDB-GENO-960809-7 | Parichy Lab derivative of AB, ABwp |
| Strain, strain background (*Danio rerio*) | Tg(krt4:lyn-EGFP)sq18 | PMID:24400120 | RRID:ZDB-TGCONSTRCT-140415-2 | |
| Strain, strain background (*Danio rerio*) | Tg(krtt1c19e:lyn-tdTomato)sq16 | PMID:24400120 | RRID:ZDB-TGCONSTRCT-140424-2 | |
| Strain, strain background (*Danio rerio*) | Tg(krt4:TetGBDTRE-v2a-cdc42DN)ir.rt3 | This paper | N/A | Transgenic line. Maintained in Eom lab. Described in Materials and methods. |
| Strain, strain background (*Danio rerio*) | Tg(krtt1c19e:TetGBDTRE-v2a-SuHDN)ir.rt18 | This paper | N/A | Transgenic line. Maintained in Eom lab. Described in Materials and methods. |
| Strain, strain background (*Danio rerio*) | Tg(Tp1:H2BmCherry)jh11 | PMID:19595765 | RRID:ZDB-TGCONSTRCT-120419-6 | |
| Strain, strain background (*Danio rerio*) | Tg(Tp1:EGFP)um13 | PMID:19595765 | RRID:ZDB-TGCONSTRCT-210311-3 | |

*Continued on next page*

*Continued*

| Reagent type (species) or resource | Designation | Source or reference | Identifiers | Additional information |
|---|---|---|---|---|
| Strain, strain background (*Danio rerio*) | Tg(krt4:nVenus-v2a-il17rd)ir.rt16 | This paper | N/A | Transgenic line. Maintained in Eom lab. Described in Materials and methods. |
| Strain, strain background (*Danio rerio*) | Tg(krt4:il17ra1a-v2a-mCherry)ir.rt17 | This paper | N/A | Transgenic line. Maintained in Eom lab. Described in Materials and methods. |
| Strain, strain background (*Danio rerio*) | clint1a^hi1520Tg/+(AB) | ZIRC | RRID:ZDB-FISH-150901-2948 | |
| Strain, strain background (*Danio rerio*) | Tg(mpeg1:il17a-v2a-mCherry)ir.rt21 | This paper | N/A | Transgenic line. Maintained in Eom lab. Described in Materials and methods. |
| Strain, strain background (*Danio rerio*) | Tg(lfng:mCherry)ir.rt20 | This paper | N/A | Transgenic line. Maintained in Eom lab. Described in Materials and methods. |
| Strain, strain background (*Danio rerio*) | il17a-/- | This paper | N/A | CRISPR-CAS9 knock-out line. Maintained in Eom lab. Described in Materials and methods. |
| Sequence-based reagent | actb1_forward | PMID:26891128 | ENSDARG00000037746 | 5'- CATCCGTAAGGACCTGTATGCCAAC- 3' |
| Sequence-based reagent | actb1_reverse | PMID:26891128 | ENSDARG00000037746 | 5'- AGGTTGGTCGTTCGTTTGAATCTC- 3' |
| Sequence-based reagent | il17rd_forward | This paper | N/A | 5'- AAATGCAGCTATAAGCAGGGA –3' |
| Sequence-based reagent | il17rd_reverse | This paper | N/A | 5'- ATGTGACTCCGAGTTTGCGA –3' |
| Sequence-based reagent | il17ra1a_forward | This paper | N/A | 5'- TGTAAGCACTGAAGCCGATGT –3' |
| Sequence-based reagent | il17ra1a_reverse | This paper | N/A | 5'- CACATCGAGGATGCGGAAGT –3' |
| Sequence-based reagent | il17a/f1_forward | This paper | N/A | 5'- GATAGACGGCGTTGAGGTCC –3' |
| Sequence-based reagent | il17a/f1_reverse | This paper | N/A | 5'- TCCACATAAGGACGAACGCA –3' |
| Sequence-based reagent | cdc42_forward | This paper | N/A | 5'- ATACGTGGAATGCTCCGCTC –3' |
| Sequence-based reagent | cdc42_reverse | This paper | N/A | 5'- ACGCTTCTTCTTGGGCTCTG –3' |
| Sequence-based reagent | rac1_forward | This paper | N/A | 5'- AGGCCATAAAGTGTGTGGTCGTC –3' |
| Sequence-based reagent | rac1_reverse | This paper | N/A | 5'- GTAGGAAAGCGGTCGAAGCCTGTC - 3' |
| Sequence-based reagent | krt4_forward | This paper | N/A | 5'- TCAACCAGGTCTATCTCTTACTCC- 3' |
| Sequence-based reagent | krt4_reverse | This paper | N/A | 5'- AACAGGCTCTGGTTGACAGTTAC- 3' |
| Sequence-based reagent | krtt1c19e_forward | This paper | N/A | 5'- GCTACCACCTTCTCCAGCGGAAG- 3' |
| Sequence-based reagent | krtt1c19e_reverse | This paper | N/A | 5'- TGCAGCACCAAATCCTCCACCAG- 3' |
| Sequence-based reagent | dlc_forward | This paper | N/A | 5'- CGGGAATCGTCTCTTTGATAAT- 3' |

*Continued on next page*

Continued

| Reagent type (species) or resource | Designation | Source or reference | Identifiers | Additional information |
|---|---|---|---|---|
| Sequence-based reagent | dlc_reverse | This paper | N/A | 5'- CTCACCGATAGCGAGTCTTCTT- 3' |
| Sequence-based reagent | notch1a_forward | This paper | N/A | 5'- CGGCATCAACACCTACAACTG- 3' |
| Sequence-based reagent | notch1a_reverse | This paper | N/A | 5'- TGGACACTCGCAGAAGAAGG- 3' |
| Sequence-based reagent | notch2_forward | This paper | N/A | 5'- GAGTGTGTGGACCCGTTAGTATG- 3' |
| Sequence-based reagent | notch2_reverse | This paper | N/A | 5'- GCAGGCATCATCAATGTGACAC- 3' |
| Sequence-based reagent | notch3_forward | This paper | N/A | 5'- TCAGGATTGTTCTCTCGTTGATG- 3' |
| Sequence-based reagent | notch3_reverse | This paper | N/A | 5'- GTGTTAAAGCATGTACCACCATTG- 3' |
| Sequence-based reagent | il17a/f1_forward | This paper | N/A | 5'- CCTCCGCTTTCTTATGGTGAGTATAGC –3' |
| Sequence-based reagent | il17a/f1_reverse | This paper | N/A | 5'- GGAACCACTGAATGCCAATATAGCAG –3' |
| Recombinant DNA reagent | krt4:LifeAct-mRuby | This paper | N/A | Assembled using Gibson Assembly |
| Recombinant DNA reagent | krt4:mCherryTuba | This paper | N/A | Assembled using Gibson Assembly |
| Recombinant DNA reagent | krt4:palmEGFP | This paper | N/A | Assembled using Gibson Assembly |
| Recombinant DNA reagent | pDestTol2-CG2(Destination Vector (#395)) | Gift. PMID:17937395 | N/A | |
| Recombinant DNA reagent | pDestTol2-exorh:mCherry (Destination Vector) | Addgene | RRID:Addgene_195989 | |
| Recombinant DNA reagent | pDestTol2-exorh:EGFP (Destination Vector) | Addgene | RRID:Addgene_195983 | |
| Commercial assay or kit | MEGAshortscript T7 High Yield Transcription Kit | Invitrogen | AM1354 | |
| Commercial assay or kit | TrueCut Cas9 Protein v2 | Invitrogen | A36497 | |
| Commercial assay or kit | LR Clonase II | Invitrogen | 12538120 | |
| Commercial assay or kit | Gibson AssemblyMaster Mix | NEB | E2611L | |
| Commercial assay or kit | Gibco Stem Pro Accutase Cell Dissociation Reagent | Thermo Fisher Scientific | A1110501 | |
| Commercial assay or kit | SuperScript III CellsDirect cDNA Synthesis Kit | Fisher Scientific | 18-080-200 | |
| Commercial assay or kit | PowerUP SYBR Green Master Mix | Thermo Fisher Scientific | A25742 | |
| Commercial assay or kit | Click-iT EdU Cell Proliferation Kit for Imaging, Alexa Fluor 594 dye | Thermo Fisher Scientific | C10339 | 500 uM |
| Chemical compound, drug | DMSO | Sigma-Aldrich | D8418 | |
| Chemical compound, drug | Doxycycline hyclate | Sigma-Aldrich | 24390-14-5 | 37.5 µM;27.5 µM |

*Continued*

| Reagent type (species) or resource | Designation | Source or reference | Identifiers | Additional information |
|---|---|---|---|---|
| Chemical compound, drug | Dexamethasone | Sigma-Aldrich | 50-02-2 | 50 μM |
| Chemical compound, drug | LY411575 | Thomas Scientific | C817J63 | 3 μM |
| Chemical compound, drug | ML141 | Sigma-Aldrich | 71203-35-5 | 2 μM |
| Chemical compound, drug | Acridine Orange | Sigma-Aldrich | 65-61-2 | 2 μg/mL |
| Chemical compound, drug | 10% Formaldehyde | LabChem | LC146602 | 3.7% |
| Chemical compound, drug | TritonX-100 | CHEM-IMPEX | 01279 | 0.5%; 0.2% |
| Chemical compound, drug | 10 x PBS Buffer | Invitrogen | AM9624 | 1 x |
| Chemical compound, drug | 4% PFA | Thermo Fisher Scientific | J19943.K2 | |
| Chemical compound, drug | Tissue-Plus O.C.T. Compound | Fisher Healthcare | 23-730-571 | |
| Chemical compound, drug | Normal Goat Serum (10%) | Thermo Fisher Scientific | 50197Z | |
| Chemical compound, drug | DAPI Fluoromount-G | SouthernBiotech | 0100–20 | |
| Software, algorithm | Fiji/ImageJ | National Institutes of Health | RRID:SCR_002285 | |
| Software, algorithm | Adobe Illustrator | Adobe Inc | RRID:SCR_010279 | |
| Software, algorithm | GraphPad Prism 9 | GraphPad | RRID:SCR_002798 | |

## Transgenesis and transgenic line production

Transgenic fish lines with cell-type specific and temporally inducible expression of human *cdc42DN* and *SuHDN* were generated following the protocol described by *Eom et al., 2015*. These constructs were driven by a 2.2 kb *krt4* promoter obtained from T. Carney for the expression in fully differentiated keratinocytes and a 3.9 kb *krtt1c19e* promoter, also from T. Carney, for the expression in undifferentiated keratinocytes and epidermal stem cells. The efficiency of gene expression in both *Tg(krt4:TetGBDTRE-v2a-cdc42DN)*[ir.rt3] and *Tg(krtt1c19e:TetGBDTRE-v2a-SuHDN)*[ir.rt18], induced by doxycycline (Dox; Sigma-Aldrich) and dexamethasone (Dex; Sigma-Aldrich), was performed in both F0 mosaic fish and non-mosaic stable lines, with consistent results. The coding sequence of *il17rd* (ZDB-GENE-020320–5) and *il17ra1a* (ZDB-GENE-070705–242) were isolated from WT(ABb) cDNA and used to generate constructs driven by the *krt4* promoter, which were assembled using Gateway assembly into the pDestTol2CG2 destination vector to produce *Tg(krt4:nVenus-v2a-il17rd)*[ir.rt16] and *Tg(krt4:il17ra1a-v2a-mCherry)*[ir.rt17]. Similarly, *Tg(mpeg1:il17a-v2a-mCherry)*[ir.rt21] was produced by isolating the coding sequence of *il17a* (ZDB-GENE-061031–3) from WT(ABb) cDNA and using Gateway assembly to assemble the construct driven by the *mpeg1* promoter into the pDestTol2-exorh:mCherry destination vector [Addgene #195989]. The *lunatic fringe* reporter line, *Tg(lfng:mCherry)*[ir.rt20], was generated by first amplifying the *lfng* promoter sequence (ZDB-GENE-980605–16) using WT gDNA and then used Gateway assembly into the pDestTol2-exorh:EGFP destination vector [Addgene #195983]. These constructs were then injected into WT(ABb) fish to generate mosaic F0 fish, which were then bred in pairs to find germline carriers to produce non-mosaic transgenic lines. Actin was examined with *LifeAct-mRuby* (Addgene #166977; *Barros-Becker et al., 2017*) driven by *krt4* promoter. Microtubules in *krt4+* keratinocytes and cytonemes were labelled using subclones from Tuba1-mCherry (Addgene #49149; *Friedman et al., 2010*).

## Generation of *il17a* mutants by the CRISPR/Cas9 System

Disruption of *il17a/f1* (GenBank accession number NM_001020787) was performed using CRISPR/Cas9 technology. The target site in exon 2 was selected using SMART and the corresponding sequence was 5'-AGCAGATCATTCATACCGGC-3'. Single guide RNA (sgRNA) was synthesized using MEGAshortscript T7 High Yield Transcription Kit (Invitrogen). Then, 1 µg/µL TrueCut Cas9 Protein v2 (Invitrogen) and 300 ng/µL sgRNA were co-injected into the one-cell stage zebrafish embryos to knockout the *il17a* gene. Founder fish (F0) were outcrossed with WT(ABb) to generate *il17a* heterozygous mutants, which were identified upon adulthood and intercrossed to obtain *il17a* homozygote mutants. The PCR primers for genotyping were as follows: Forward 5'- CCTCCGCTTTCTTATGGTGA GTATAGC –3' and Reverse 5'- GGAACCACTGAATGCCAATATAGCAG –3'.

## Drug treatments

Zebrafish were kept in E3 medium with or without drugs (Dox/Dex) during both the light and the dark cycles for long-term drug treatment experiments. For overnight time-lapse images, fish subjected to drug treatment were transferred to fresh E3 medium 30 min before explant preparation to minimize background fluorescence from the drugs. Zebrafish with *krt4:palmEGFP* injected TetGBD-containing transgenics and non-transgenic siblings were treated with 37.5 µM Dox and 50 µM Dex one day prior to time-lapse imaging, while fish for long-term (3 day) drug treatment received 27.5 µM Dox and 50 µM Dex starting from SSL7.0.

Notch inhibitor, LY411575 (10 mM; Thomas Scientific) was prepared in DMSO (Sigma-Aldrich), and fish were treated with drug [LY411575 (3 µM)] or vehicle from SSL7.0. For acute drug administration, Cdc42 inhibitor ML141 (Sigma-Aldrich), prepared in DMSO, was added to fish medium [ML141 (2 µM)] 30 min prior to explant preparation for time-lapse imaging.

## Acridine Orange cell death assay

Acridine Orange (5 mg/mL; Sigma-Aldrich) was prepared in Invitrogen UltraPure Distilled Water. Fish were washed in 1 x E3 medium 30 min before being transferred to 1 x E3 medium containing AO (2 µg/mL). The fish were incubated in AO for 30 min and then rinsed in regular 1 x E3 medium three times for 10 min each to minimize background fluorescence during imaging. Dead cells, pseudo-colored in magenta, were counted for all groups.

## Time-lapse and still imaging

Ex vivo time-lapse imaging of zebrafish epidermal cells within their native tissue environment was acquired at 3 min intervals for 10 hr. This was achieved using a 40 x water-immersion objective on a Leica TCS SP8 confocal microscope equipped with a resonant scanner and two HyD detectors. Larvae were imaged at SSL7.5, unless otherwise specified. For still images, the region directly underneath the dorsal fin of the zebrafish was captured, with a total of 10 tiles per fish.

## Cytoneme analysis

In zebrafish keratinocytes, lamellipodial extensions are the dominant extension type, and most filopodial extensions are less than 1 µm in length; both are not easily visible at the confocal resolution we used for this study. Thus, it is easy to distinguish filopodia from cytonemes, as cytonemes have a minimum length of 4.36 µm in our observations. We did not use the width parameter since there are no other protrusions except cytonemes. We calculated the cytoneme extension frequency by counting how many cytonemes extended from a cell per hour. We analyzed movies with 3 min intervals over a total of 10 hr, as described in the section above.

## RT-PCR, qRT-PCR, and FACS

Zebrafish at SSL7.5 were skinned (N=15 per group), and the dissected tissues were placed in cold 1 x PBS. Tissues were washed by pipetting up and down and then centrifuged to remove the supernatant. Next, 1 mL Gibco Stem Pro Accutase Cell Dissociation Reagent was added to resuspend tissues, and the tube was incubated for at least 10 min at 37°C to dissociate the cells. After incubation, the cells were passed through 40 µm cell strainers to remove large tissue chunks before subjecting them to FACS (Fluorescence-activated cell sorting). FACS enabled the separation and collection of desired cell populations based on different cell membrane markers. EGFP+ and tdTomato+ cells were

collected for downstream non-quantitative RT-PCR and quantitative RT-PCR. For cDNA synthesis, the SuperScript III CellsDirect cDNA Synthesis Kit was employed. Quantitative PCR was conducted on an Applied Biosystems QuantStudio 3 Real-Time PCR Instrument using PowerUP SYBR Green Master Mix and custom primers. *actb1*: 5'- CATCCGTAAGGACCTGTATGCCAAC- 3', 5'- AGGTTGGTCGTT CGTTTGAATCTC- 3' (*Hu et al., 2016*); *il17rd*: 5'- AAATGCAGCTATAAGCAGGGA –3', 5'- ATGTGACT CCGAGTTTGCGA –3'; *il17ra1a*: 5'- TGTAAGCACTGAAGCCGATGT –3', 5'- CACATCGAGGATGCGG AAGT –3'; *il17a/f1*: 5'- GATAGACGGCGTTGAGGTCC –3', 5'- TCCACATAAGGACGAACGCA –3'; *cdc42*: 5'- ATACGTGGAATGCTCCGCTC –3', 5'- ACGCTTCTTCTTGGGCTCTG –3';; *rac1*: 5'- AGGC CATAAAGTGTGTGGTCGTC –3', 5'- GTAGGAAAGCGGTCGAAGCCTGTC –3'. Quantitative PCRs were run with at least triplicate biological and technical replication for each sample.

Non-quantitative RT-PCR amplifications were performed with 40 cycles (*actb1, dlc, krt4, krtt1c19e, notch1a, notch2, notch3*) using Takara PrimeStar GXL DNA Polymerase. *actb1*: 5'- CATCCGTAAGGA CCTGTATGCCAAC- 3', 5'- AGGTTGGTCGTTCGTTTGAATCTC- 3' (*Hu et al., 2016*); *krt4*: 5'- TCAA CCAGGTCTATCTCTTACTCC- 3', 5'- AACAGGCTCTGGTTGACAGTTAC- 3'; *krtt1c19e*: 5'- GCTA CCACCTTCTCCAGCGGAAG- 3', 5'- TGCAGCACCAAATCCTCCACCAG- 3'; *dlc*: 5'- CGGGAATC GTCTCTTTGATAAT- 3', 5'- CTCACCGATAGCGAGTCTTCTT- 3'; *notch1a*: 5'- CGGCATCAACACCTAC AACTG- 3', 5'- TGGACACTCGCAGAAGAAGG- 3'; *notch2*: 5'- GAGTGTGTGGACCCGTTAGTATG- 3', 5'- GCAGGCATCATCAATGTGACAC- 3'; *notch3*: 5'- TCAGGATTGTTCTCTCGTTGATG- 3', 5'- GTGTTAAAGCATGTACCACCATTG- 3'.

## Cell counts

For all experiments, 6 out of 10 tiles of still images, which represent the central trunk area directly beneath the dorsal fin, were selected for cell counts. To ensure consistency across all samples and minimize discrepancies resulting from variations in fish trunk widths, a region of interest (ROI), 290μm x 200μm per tile, was created using the rectangle selection tool in ImageJ. Z-stack imaging was confined to the superficial/periderm layer based on the cross-sectional view, and the final count included both *krt4+* and *krtt1c19e+* cells at the periderm layer.

## Intensity analyses

To analyze Notch expression level, tile images were merged separately for each fish. Merged images were then sectioned in ImageJ using the polygon selection tool to encompass only the fish trunk, and maximal projection was utilized to obtain the mean gray value. For *krtt1c19e+* expression in the periderm layer, the z-stack was set as previously described, and the mean gray value of *krtt1c19e+* expression was obtained.

## Microridge length analysis

The workflow for microridge length analysis was modified from *van Loon et al., 2020*. In brief, raw images obtained from the Leica TCS SP8 confocal microscope were imported into ImageJ with z-stacks adjusted to include only the cell of interest. The cell outline was traced by the polygon tool, and the area around the cell was cleared. Then the segmented cell was converted into grayscale followed by automatic adjustment of Brightness and Contrast. Images were then sharpened once, and a Gaussian Blur filter was applied with a 0.5 Sigma (Radius) before converting into a binary format for skeletonization. Branch lengths were calculated by the Analyze Skeleton (2D/3D) feature.

## Western blotting

Total protein was obtained from adult zebrafish tissues with the protocol modified from *Xue and Corti, 2022* In brief, tissues were homogenized in 200 μL cold NP-40 Lysis buffer (BP-119X; Boston BioProducts, Inc) with 1 x protease inhibitor cocktail (539131; Sigma-Aldrich). The homogenates were incubated on ice for 30 min before centrifuging at 14,000 x *g* at 4°C for 30 min. 5 μL of the supernatants were used to quantify total protein with the BCA protein assay. The supernatants were then subjected to SDS-PAGE and transferred to PVDF membranes. The primary antibody used was Rabbit anti-IL-17 a/f1 at 1:1000 (KP1239Z-100; KINGFISHER BIOTECH, INC) and Rabbit anti-β-actin at 1:5000 (GTX637675; GeneTex). The secondary antibody used was anti-rabbit IgG, HRP-linked Antibody #7074; Cell Signaling TECHNOLOGY. The experiments were performed in triplicates and representative results are shown.

## 5-Ethynnyl-2'-deoxyuridine (EdU) labeling

The Click-iT EdU Cell Proliferation Kit (Thermo Fisher) was used to label DNA of proliferating cells with AlexaFluor-594. The protocol was modified for zebrafish as follows: zebrafish larvae with desired transgenic backgrounds were raised to SSL7.5 or treated with drugs as described earlier and incubated in EdU solution (500 µM in E3) at 28°C for at least 2 hr. Larvae were fixed in 3.7% formaldehyde in PBS overnight, then washed and transferred to permeabilization solution, 0.5% Triton X-100 in PBS for 1 hr. Permeabilized larvae were then treated according to manufacturer protocol. Proliferating cells were defined as cells that are positive for EdU.

## Immunohistochemistry

The protocol was modified from Abcam IHC-Frozen protocols. Zebrafish larvae were raised to desired developmental stages. A fresh transplant was prepared from the trunk and frozen in O.C.T. Tissues were sectioned according to the protocol and immediately fixed with 4% PFA for 10 min at room temperature, then washed with 1 x PBS. Samples were permeabilized in 0.2% Triton in 1 x PBS for 10 min and blocked in 10% Normal Goat Serum for 1 hr at room temperature. The primary antibody used was Rabbit anti-Hes1 (MA5-32258; Invitrogen). Primary antibodies were diluted in 10% Normal Goat serum (50197Z; Thermo Fisher) 1:250 and samples were incubated overnight at 4°C. Samples were washed with 1 x PBS. The secondary antibody used was Donkey Anti-Rabbit IgG H&L Alexa Fluor 488 (ab150073; Abcam). Secondary antibodies were diluted 1:250 in 10% Normal Goat Serum and samples were incubated for 1 hr at room temperature and washed with fresh 1 x PBS. DAPI Fluoromount-G (0100–20; SouthernBiotech) was used to mount coverslip and prepared according to manufacturer protocol.

## Quantification and statistical analysis

GraphPad Prism software version 9.0.0 for Windows (GraphPad Software, San Diego, CA, USA) was used to perform statistical analyses. Continuous data were evaluated by Student t-test and Post hoc means were compared by Tukey-Kramer HSD.

## Acknowledgements

We thank Drs. Plikus, Andersen and Parsons for the invaluable discussions, and all the members of Eom lab for maintaining our fish lines. This study was supported by the National Institutes of Health grant R35GM142791 (to DSE) and National Institutes of Health T32 training grant AR080622 (to YW).

## Additional information

### Funding

| Funder | Grant reference number | Author |
|---|---|---|
| National Institute of Arthritis and Musculoskeletal and Skin Diseases | T32AR080622 | Yi Wang |
| National Institute of General Medical Sciences | R35GM142791 | Dae Seok Eom |

The funders had no role in study design, data collection and interpretation, or the decision to submit the work for publication.

### Author contributions

Yi Wang, Conceptualization, Formal analysis, Validation, Investigation, Visualization, Methodology, Writing – original draft, Writing – review and editing; Thomas Nguyen, Formal analysis, Validation, Investigation, Visualization, Methodology; Qingan He, Oliver Has, Kiarash Forouzesh, Formal analysis, Methodology; Dae Seok Eom, Conceptualization, Resources, Formal analysis, Supervision, Funding

acquisition, Validation, Investigation, Visualization, Methodology, Writing – original draft, Project administration, Writing – review and editing

### Author ORCIDs
Yi Wang (ORCID) https://orcid.org/0000-0001-7409-4335
Thomas Nguyen (ORCID) http://orcid.org/0009-0006-3362-3468
Oliver Has (ORCID) http://orcid.org/0009-0000-5085-7211
Kiarash Forouzesh (ORCID) http://orcid.org/0009-0005-1850-7808
Dae Seok Eom (ORCID) https://orcid.org/0000-0002-0617-8788

### Ethics
All animal work in this study was conducted with the approval of the University of California Irvine Institutional Animal Care and Use Committee (Protocol #AUP-25-002) in accordance with institutional and federal guidelines for the ethical use of animals.

Reviewer #1 (Public review): https://doi.org/10.7554/eLife.97400.3.sa1
Reviewer #2 (Public review): https://doi.org/10.7554/eLife.97400.3.sa2
Author response https://doi.org/10.7554/eLife.97400.3.sa3

---

## Additional files

### Supplementary files
MDAR checklist

### Data availability
All data generated or analyzed during this study are available upon request from Dae Seok Eom ( dseom@uci.edu). Due to the large size (approximately 3TB) of high-resolution time-lapse data, it is not feasible to upload to public data repositories. Researchers may request raw or processed data without restrictions, provided they cite the source when used. Maximum intensity projection or tiled images were generated using LasX (Leica) or ImageJ software, utilizing the functionality provided by these programs.

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
