## [Editor Report · eLife Assessment]

This is a **valuable** study showing that differentiated cells of the zebrafish skin form membrane protrusions called cytonemes that contact and likely transmit Notch signals to cells of the undifferentiated layer below. The data are **convincing** that cytoneme like protrusions from the periderm are required for proper periderm structure, proliferation, gene expression, and Notch signaling. Evidence that inflammatory signaling through IL-17 affects epidermal differentiation, Notch and cytoneme formation is **solid**, but whether these are through a single common or two parallel pathways requires further investigation.

---

## [Referee Report · Reviewer #1 (Public review)]

Summary:

In this paper, Wang et al show that differentiated peridermal cells of the zebrafish epidermis extend cytoneme-like protrusions toward the less differentiated, intermediate layer below. They present evidence that expression of a dominant-negative cdc42, inhibits cytoneme formation and leads to elevated expression of a marker of undifferentiated keratinocytes, krtt1c19e, in the periderm layer. It is demonstrated that Delta-Notch signaling is involved in keratinocyte differentiation and that loss of cytonemes correlates with a loss of Notch signaling. Finally, changes in expression of the inflammatory cytokine IL-17 and its receptors is shown to affect cytoneme number and periderm structure in a manner similar to Notch and cdc42 perturbations.

Strengths:

Overall, the idea that differentiated cells signal to underlying undifferentiated cells via membrane protrusions in skin keratinocytes is interesting and novel, and it is clear that periderm cells send out thin membrane protrusions that contain a Notch ligand. Further, and perturbations that affect cytoneme number, Notch signaling and IL-17 expression clearly lead to changes in periderm structure and gene expression.

Weaknesses:

The mechanisms by which IL-17 affects cytoneme formation requires further investigation.

---

## [Referee Report · Reviewer #2 (Public review)]

Summary:

The aim of the study was to understand how cells of the skin communicate across dermal layers. The research group has previously demonstrated that cellular connections called airinemes contribute to this communication. The current work builds upon this knowledge by showing that differentiated keratinocytes also use cytonemes, specialized signaling filopodia, to communicate with undifferentiated keratinocytes. They show that cytonemes are the more abundant type of cellular extension used for communication between the differentiated keratinocyte layer and the undifferentiated keratinocytes. Disruption of cytoneme formation led to expansion of the undifferentiated keratinocytes into the periderm, mimicking skin diseases like psoriasis. The authors go on to show that disruption of cytonemes results in perturbations in Notch signaling between the differentiated keratinocytes of the periderm and the underlying proliferating undifferentiated keratinocytes. Further the authors show that Interleukin-17, also known to drive psoriasis, can restrict formation of periderm cytonemes, possibly through the inhibition of Cdc42 expression. This work suggests that cytoneme mediated Notch signaling plays a central role in normal epidermal regulation. The authors propose that disruption of cytoneme function may be an underlying cause of various human skin diseases.

Strengths:

The authors provide strong evidence that periderm keratinocytes cytonemes contain the notch ligand DeltaC to promote Notch activation in the underlying intermediate layer to regulate accurate epidermal maintenance.

Weaknesses:

The impact of the study would be increased if the mechanism by which Interlukin-17 and Cdc42 collaborate to regulate cytonemes was defined. Experiments measuring Cdc42 activity, rather than just measuring expression, would strengthen the conclusions.

Comments on revisions:

The authors have sufficiently addressed my critiques from the initial round of evaluation. They have included useful representative images, clarified how they scored cytonemes and provided additional controls/experimental conditions that improve the rigor of the study. The results provided now support the key conclusions of the study.

---

## [Author Response]

The following is the authors’ response to the original reviews

**Public Reviews:**

**Reviewer #1 (Public Review):**
Summary:In this paper, Wang et al show that differentiated peridermal cells of the zebrafish epidermis extend cytoneme-like protrusions toward the less differentiated, intermediate layer below. They present evidence that expression of a dominant-negative cdc42, inhibits cytoneme formation and leads to elevated expression of a marker of undifferentiated keratinocytes, krtt1c19e, in the periderm layer. Data is presented suggesting the involvement of Delta-Notch signaling in keratinocyte differentiation. Finally, changes in expression of the inflammatory cytokine IL-17 and its receptors is shown to affect cytoneme number and periderm structure in a manner similar to Notch and cdc42 perturbations.Strengths:Overall, the idea that differentiated cells signal to underlying undifferentiated cells via membrane protrusions in skin keratinocytes is interesting and novel, and it is clear that periderm cells send out thin membrane protrusions that contain a Notch ligand. Further, perturbations that affect cytoneme number, Notch signaling, and IL-17 expression clearly lead to changes in periderm structure and gene expression.Weaknesses:More work is needed to determine whether the effects on keratinocyte differentiation are due to a loss of cytonemes themselves, or to broader effects of inhibiting cdc42. Moreover, more evidence is needed to support the claim that periderm cytonemes deliver Delta ligands to induce Notch signaling below. Without these aspects of the study being solidified, understanding how IL-17 affects these processes seems premature.
**Reviewer #2 (Public Review):**
Summary:The aim of the study was to understand how cells of the skin communicate across dermal layers. The research group has previously demonstrated that cellular connections called airinemes contribute to this communication. The current work builds upon this knowledge by showing that differentiated keratinocytes also use cytonemes, specialized signaling filopodia, to communicate with undifferentiated keratinocytes. They show that cytonemes are the more abundant type of cellular extension used for communication between the differentiated keratinocyte layer and the undifferentiated keratinocytes. Disruption of cytoneme formation led to the expansion of the undifferentiated keratinocytes into the periderm, mimicking skin diseases like psoriasis. The authors go on to show that disruption of cytonemes results in perturbations in Notch signaling between the differentiated keratinocytes of the periderm and the underlying proliferating undifferentiated keratinocytes. Further, the authors show that Interleukin-17, also known to drive psoriasis, can restrict the formation of periderm cytonemes, possibly through the inhibition of Cdc42 expression. This work suggests that cytoneme-mediated Notch signaling plays a central role in normal epidermal regulation. The authors propose that disruption of cytoneme function may be an underlying cause of various human skin diseases.Strengths:The authors provide strong evidence that periderm keratinocytes cytonemes contain the notch ligand DeltaC to promote Notch activation in the underlying intermediate layer to regulate accurate epidermal maintenance.Weaknesses:The impact of the study would be increased if the mechanism by which Interlukin-17 and Cdc42 collaborate to regulate cytonemes was defined. Experiments measuring Cdc42 activity, rather than just measuring expression, would strengthen the conclusions.
**Reviewer #3 (Public Review):**
Summary:Leveraging zebra fish as a research model, Wang et al identified "cytoneme-like structures" as a mechanism for mediating cell-cell communications among skin epidermal cells. The authors further demonstrated that the "cytoneme-like structures" can mediate Notch signaling, and the "cytoneme-like structures" are influenced by IL17 signaling.Strengths:Elegant zebrafish genetics, reporters, and live imaging.Weaknesses: (minor)This paper focused on characterizing the "cytoneme-like structures" between different layers and the NOTCH signaling. However, these "cytoneme-like structures" observed in undifferentiated KC (Figure 2B), although at a slightly lower frequency, were not interpreted. In addition, it is unclear if these "cytoneme-like structures" can mediate other signaling pathways than NOTCH.

We are currently investigating the role of cytoneme-like protrusions extended from undifferentiated keratinocytes and their role is still under investigation. We believe that addressing the function of undifferentiated keratinocyte cytonemes and exploring whether peridermal cytoneme can mediate other signaling pathways is beyond the scope of the current manuscript. However, we hope to publish our discoveries about them soon. It is worth noting that cytonemes mediate other morphogenetic signals, such as Hh, Wnt, Fgf, and TGFbeta in other contexts.

Overall, this is a solid paper with convincing data reporting the "cytoneme-like structures" in vivo, and with compelling data demonstrating the roles in NOTCH signaling and the regulation by IL17.

These findings provide a foundation for future work exploring the "cytoneme-like structures" in the mammalian system and other epithelial tissue types. This paper also suggests a potential connection between the "cytoneme-like structures" and psoriasis, which needs to be further explored in clinical samples.

**Recommendations for the authors:**

**Reviewer #1 (Recommendations For The Authors):**
Major points- In general, representative images from each experiment should accompany the graphs shown. The inclusion of still frames from time-lapse imaging experiments in the main figures would help the reader understand the morphology and dynamics of these protrusions in control, cdc42, and IL-17 manipulations.

Thank you for the comments. We appreciate your suggestion to include representative images alongside the graphs to better illustrate the morphology and dynamics of these protrusions.

In response, we have made the following additions to our main figures.

Figure 3A now includes still images from time-lapse movies for both control and cdc42 manipulations.

Figure 5A and 6A,C now include still images for il17 manipulations.

- Data in Figure 3 is crucial as it demonstrates that cdc42DN selectively impairs cytoneme extensions without affecting other actin-based structures. It also shows that cdc42DN leads to upregulation of krtt1c19e in periderm. Therefore, these data should be presented in a comprehensive way. Still, frames of high mag views of time-lapse images from control and cdc42DN should be included in the figure. Similarly, a counter label (E-Cadherin, perhaps) showing the presence of all three layers and goblet cells at different focal planes capturing the different layers of the skin should be included. It is stated that the goblet cell number is unaffected, but they seem to be absent in the image shown in Figure 3B.

In this revised version, we have included magnified cross-sectional views. In addition to the images of the peridermal layer from the original version, we have now included the underlying intermediate and basal stem cell layers (Figure 3C-C”). We hope these data convincingly show that peridermal keratinocytes in cytoneme inhibited animals co-express krt4 and krtt1c19e markers, suggesting that peridermal keratinocytes are not fully differentiated.

We agree that the goblet cells in this particular image of experimental group appear largely absent, however, as we quantified many animals, the number of goblet cells was not significantly different between controls and experimental (Figure S2).

- The effects on periderm architecture upon broad cdc42 inhibition may not be directly due to a loss of cytonemes. Performing this experiment in a mosaic manner to determine if the effects are local and in the range of cytoneme protrusion would strengthen the conclusions. Adding a secondary perturbation to inhibit cytoneme formation in periderm cells would also strengthen the conclusions that defects are not related specifically to cdc42 inhibition, but cytonemes themselves.

Thank you for the suggestion. We confirmed that mosaic expression of cdc42DN in peridermal keratinocytes elicited local disorganization, and elevated *krtt1c19e* expression as we seen in transgenic lines. Also, the cdc42DN expressing cells exhibited significantly lower cytoneme extension frequency.

In addition, we found that like cdc42DN, rac1DN expressing keratinocytes exhibited significant decrease in cytoneme extension frequency, but rhoabDN show no effects (new Figure S3). These data suggest that cytoneme extension is regulated by cdc42 and rac1 but not rhoab. Further investigation is required however, at least these data suggest that the effects we observe is likely the loss of cytonemes not just specifically to cdc42 inhibition.

- Figure 4. The inclusion of an endogenous reporter of Notch activity, like Hes or Hey immunofluorescence, would strengthen the conclusion that the intermediate layer is Notch responsive.

Thank you for the suggestion. In this revised version, we have included immunostaining data in Figure 4D demonstrating that Her6 (the orthologous to human HES1) protein is expressed in the intermediate layer.

- It is not clear where along a differentiation trajectory Notch signaling and cytonemes are needed. What happens to the intermediate layer when Notch signaling or cdc42 is inhibited? Do the cells become more basal-like? Or failing to become periderm? Meaning - is Notch promoting the basal to intermediate fate transition, or the intermediate to periderm transition? A more comprehensive characterization of basal, intermediate, and periderm differentiation with markers selective to each layer would help define which step in the process is being altered.

Notch signaling is known to regulate keratinocyte terminal differentiation. Thus, it requires in the process from intermediate to peridermal transition. We observed peridermal keratinocytes still strongly express krt19 suggesting their terminal differentiation is inhibited when cytoneme mediated Notch signaling is compromised.

As seen on Figure 3C”, peridermal keratinocytes express both krt4 and krtt1c19e markers and they are located at the peridermal layer suggesting that they are not fully differentiated keratinocytes. As we included the images of intermediate and basal layers, we do not observe any noticeable defects in basal stem cells or complete depletion of intermediate keratinocytes (Fig 3C-C”). These observations suggest that notch signaling, activated by cytonemes, is required for the differentiation of undifferentiated intermediate keratinocytes into peridermal keratinocytes.

We included this interpretation in the main text.

- A number of times in the text it is suggested that cytonemes, Notch, and IL-17 signaling are essential for keratinocyte differentiation and proliferation, but proliferation (% cells in S-phase and M-phase) is not measured. Also, #of keratinocytes @ periderm is not an accurate way to report the number of cells in the periderm unless every cell in the larvae has been counted. It should be # cells/unit area.

In this revised version, we confirmed that the number of Edu+ cells among peridermal keratinocytes are significantly increased when cytonemes are inhibited (Figure 3F-G). Also, as indicated in the methods section, we indeed counted the cells in 290um x 200um square. We believe both of the data sufficiently suggest that the number of keratinocytes in periderm is significantly increased due to the lack of proper cytoneme mediated signaling.

- If the model is correct that Delta ligands from the periderm signal to intermediate cells to promote their differentiation and inhibit their proliferation, then depletion of Delta from Krt4 expressing cells should recapitulate the periderm phenotype.

It is a great suggestion. However, zebrafish skin express multiple delta ligands and we do not know what specific combination of Deltas are delivered via cytonemes. In this manuscript we identified Dlc is expressed along the cytonemes and krt4+ cells (revised Figure S4), however we are unsure whether other Delta ligands involve the notch activation. However, cytoneme inhibition is performed specifically in krt4+ cells and the downregulation of Notch activation are observed in krtt1c19e+ undifferentiated keratinocytes. In this revised version, we found that a Notch responsive protein Her6 is exclusively expressed in the cytoneme target keratinocytes, and cytoneme extending cells (krt4+) do not express Notch receptors.

- rtPCR data in Figure S3 is not properly controlled. Each gene should be tested in both krt4 and krtt1c19e expressing cells to determine their relative expression levels in different skin layers that are proposed to signal to one another. Are Notch ligands present in basal cells? These could be activating Notch in the intermediate layer.

Our intention was to merely confirm the Notch signaling components are expressed in cytoneme extending and receiving cells. Based on the new panel of RT-PCRs for notch signaling components, we confirmed again that dlc is expressed in cytoneme extending cells but not in receiving cells. Basal cells are also krtt1c19e+ but we did not detect dlc from them. Interestingly, we found that notch 2 is exclusively expressed in krtt1c19e+ cells but not from krt4+ cytoneme extending cells (now new Figure S4).

- It is not intuitive why NICD (activation) and SuHDN (inhibition) of Notch signaling should result in a similar effect on the periderm. What is the effect of NICD expression on the TP1:H2BGFP reporter? Does it hyperactivate as expected?

We agree reviewer’s concerns. It is well studied that *psoriasis* patients exhibits either loss or gain of notch signaling (Ota et al., 2014 Acta Histochecm Cytochem, Abdou et al., 2012 Annals of Diagnostic Pathology). However, it remains unknown the underlying mechanisms. We merely intended to showcase our zebrafish experimental manipulations recapitulate human patients’ case. However, we believe this data doesn’t require for drawing the overall conclusion but need further investigation to explain it. Thus, if the reviewers agree we want to omit it in this manuscript and leave it for future studies.

- Due to the involvement of immune signaling in hyperproliferative skin diseases the paper then investigates the role of IL-17 on cytoneme formation by overexpressing two IL-17 receptors in the periderm. Fewer cytonemes were present in the receptor over-expressing periderm cells. The rationale for overexpressing the receptors was unclear. If relevant to endogenous cytokine signaling, the periderm would be expected to express IL-17 receptors normally and respond to elevated levels of IL-17.

The rationale behind the reason of why we overexpress the IL-17 receptors is to test its autonomy of krt4+ peridermal cells. There is a debate that whether the onset of *psoriasis* is autonomous to keratinocytes or non-autonomous effects of immune malfunction. In addition to the overexpression of IL-17 receptors, we showed that the IL-17 ligand overexpression shows the sample effects on cytoneme extension (Fig. 6A-B).

- Experiments overexpressing IL-17 in macrophages are also suggested to limit cytoneme number whereas heterozygous deletion elevates them. Representative images and movies should be included to support the data. Western blots or immunofluorescence showing that IL-17 and its receptors are indeed overexpressed in the relevant layers/cell types should also be included as controls. Knockout of IL-17 protein in the new Crispr deletion mutant should also be shown.

In response to the reviewer’s comments, we have included representative images of peridermal keratinocytes in IL-17 ligand overexpressed and il17 CRISPR KO animals (Fig. 6A,C).

We have confirmed the overexpression of Il17rd, Il17ra1a and Il17a in the transgenic animals. For the il17 receptors, we FACS-sorted differentiated keratinocytes and performed qRT-PCR. Similarly, for the il17 ligand, we isolated skin tissue and conducted qRT-PCR (new Figure S7).

Additionally, we confirmed that IL-17 protein expression is undetectable in il17a CRISPR KO fish (Fig. S8C).

- Evidence that the effect of IL-17 upregulation on periderm architecture is via cytonemes is suggestive but not conclusive. Can the phenotype be rescued by a constitutively active cdc42?

We appreciate the reviewer’s suggestion. We are unsure whether constitutively active cdc42 expression can rescue IL-17 overexpression mediated reduction of cytoneme extension frequency. It is well expected that cdc42CA will stabilize actin polymerization in turn more cytonemes. However, it is also known sustained cdc42 activation can paradoxically lead to actin depolymerization. Thus, we concern it will be likely uninterpretable. Also, we need to generate a new transgenic line for this experiment and the baseline control experiments and validations take substantial amount of time and efforts with no confidence.

We and others believe that the cdc42 is a final effector molecule to regulate cytoneme extension given its role in actin polymerization. we provided the evidence that IL-17 overexpression significantly reduced cdc42 and rac1 expression (Figure 6E) and co-manipulation with IL17 overexpression and cdc42DN led to further down-regulation of cytoneme extension frequency in peridermal keratinocytes (Figure 6H).

- In a final experiment, the authors mutate a psoriasis-associated gene, clint1a gene and show an effect on cytonemes, Notch output, and periderm structure. More information about what this gene encodes, where the mRNA is expressed, and where the cell the protein should localize would help place this result in context for the reader.

In this revised manuscript we included more information about the clint1.

“The clathrin interactor 1 (*clint1*), also referred to as enthoprotin and epsinR functions as an adaptor molecule that binds SNARE proteins and play a role in clathrin-mediated vasicular transport (Wasiak, 2002). It has also been reported that *clint1* is expressed in epidermis and play an important role in epidermal homeostasis and development in zebrafish (Dodd et al., 2009)”.

Minor points- The architecture of zebrafish skin is notably distinct from that of humans and other mammals and whether parallels can be drawn with regards to cytoneme mediated signaling requires further investigation. For this reason, I believe the title should include the words 'in zebrafish skin'.

In this version, we changed the title as ‘Cytoneme-mediated intercellular signaling in keratinocytes essential for epidermal remodeling in zebrafish’.

- More details about the timing of cdc42 inhibition should be given in the main text to interpret the data. How many hours of days are the larvae treated? How does this compare to the rate of division and differentiation in the zebrafish larval epidermis?

We apologize for omitting the detailed experimental conditions for cytoneme inhibition. We have revised the main text as follows “Although the cytoneme inhibition is evident after overnight treatment with the inducing drugs, noticeable epidermal phenotypes begin to appear after 3 days of treatment. This reflects the higher cytoneme extension frequency and their potential role during metamorphic stages, which takes a couple of weeks (Figure 1C)”

- What are the genotypes of animals in Figure 4B where 'Notch expression' is being measured upon cdc42DN inhibition? Is this the TP1:H2B-GFP reporter? Again, details of the timing of this experiment are needed to evaluate the results.

We indicated the reference supplement figure for the Notch activity measure in the figure legend S4. And we added the following sentence in the main text. “Similar to the effects on the epidermis after cytoneme inhibition (Figure 3), it takes 3 days to observe a significantly reduction in Notch signal in the undifferentiated keratinocytes.”

**Reviewer #2 (Recommendations For The Authors):**
- Figure 2B: the authors indicate that the undifferentiated keratinocytes (krtt1c19e+) do extend some cytonemes. Although this behavior is not a focus of the study, it would be helpful to see an image of krtt1c19e:lyn-tdTomato cytonemes. The discussion ends with an interesting statement about downward pointed protrusions coming off the undifferentiated keratinocytes. A representative image of this should be included in Figure 2.

In this revised version, we included an image of krtt1c19e positive cell that extend cytonemes in Figure 2C.

- The evidence for hyperproliferation of the undifferentiated keratinocytes would be strengthened by quantifying proliferation. Most experiments result in increased expression of krtt1c19e in the periderm layer, but it is unclear whether this is invasion, remodeling, or incomplete differentiation of the cells. Notch suppression with krtt1c19e:SuHDN and overactivation with krtt1c19e:NICD phenocopy each other. Are there differences in proliferation vs differentiation rates in these two genotypes that result in a similar phenotype?

We appreciate the reviewer’s comments. In response to the feedback, we included Edu experiments that show increased cell proliferation in keratinocytes in periderm in experimental groups. Additionally, we observed co-expressed of both differentiated marker krt4 and undifferentiated marker krtt1c19e in the keratinocytes in periderm. Since we did not observe depletion of intermediate layer, we believe it is reasonable to conclude that the phenotype represents incomplete differentiation (new Figure 3). For the krtt1c19e:NICD question, please refer to our response to reviewer #1’ comment.

- Do Cdc42DN and il17rd or il17ra1a work in parallel or in a hierarchy of signaling events to regulate cytoneme formation?

Cdc42 is widely recognized as a final effector in cytoneme extension, given its well-established role in actin polymerization, which is critical for cytoneme extension. Our data support a model where il17 signaling acts upstream of cdc42. We showed that the overexpression of il17rd or il17ra1a significantly reduced the expression of Cdc42 (Figure 6E). In double transgenic fish overexpressing il17rd and cdc42DN, we observed a more marked decrease in cytoneme extension compared to single transgenic (Figure 6H). These results collectively indicate that, at least partially, Cdc42 functions downstream of il17 signaling in the context of cytoneme formation. However, we acknowledge that additional regulatory mechanisms may be involved, given the complexity of cellular signaling networks.

- Figure 6C: Are the effects of overexpression of il17rd specific to Cdc42, or are other Rho family GTPases like Rac and Rho also affected? Is the microridge defect (Figure 6D) also present in Tg(krt4:TetGBDTRE-v2a-cdc42DN) when induced, or could this be regulated by Rho/Rac?

We used the microridge formation as a readout to evaluate the effects of il17receptor overexpression on actin polymerization. In this revision, we demonstrate that the expression of other small GTPases is also decreased in il17rd or il17ra1a overexpressed keratinocytes (Figure 6E). Also, we confirmed that microridges exhibit significantly shorter branch length when cdc42DN or rac1DN is overexpressed (new Figure S9). It is note that we have shown that the effects on cytonemes are regulated by cdc42 and rac1 (new Figure S3).

- Please change the color of the individual data points from black to grey or another color so readers may better visualize the mean and error bars.

We agree with this comment, and in response, we have revised the figures by changing the color of the individual data points to empty circles and now the error bars are better visualized.

- Figure 1: What were the parameters used to identify an extension as a cytoneme? Please include the minimal length and max-width used in the analysis in the methods.

Thank you for the comments. We have now included the method of how we defined cytonemes and measured as follows. In zebrafish keratinocytes, lamellipodial extensions are the dominant extension type, and most filopodial extensions are less than 1µm in length, both are not easily visible at the confocal resolution we used for this study. Thus, it is easy to distinguish filopodia from cytonemes, as cytonemes have a minimum length of 4.36µm in our observations. We did not use the width parameter since there are no other protrusions except cytonemes. We calculated the cytoneme extension frequency by counting how many cytonemes extended from a cell per hour. We analyzed movies with 3-minute intervals over a total of 10 hours, as described in the section above.

- Line 149-150, (Figure S1) ML141 is a Cdc42 inhibitor, please correct the wording. Would the use of an actin polymerization inhibitor like Cytochalasin B or a depolymerizing agent (Latrunculin) increase the reduction in cytoneme formation?

Thank you for pointing it out. We have revised it in this version. We have tried Cytochalasin B or Latrunculin and the treatments killed the animals.

- Figure 2: What is the depth of the Z-axis images? Does the scale bar apply to the cross-sectional images as well? It may be beneficial to readers to expand the Z scale of the cross-section images for Figure 2C.

Sure, we enlarged the cross-sectional images. Yes, the scale bar should apply to the cross-sectional images.

- Figure 3B-B' cross-section images should be added to confirm images shown represent the periderm layer. Are there folds in the epidermis due to cdc42DN expression or are differentiated keratinocytes absent?

In response, we have included z-stack images in the revised figure 3. We found that the epidermal tissue is not flat as compared to controls, presumably due to broad cdc42DN expression (Figure 3C”).

- Figure S3: Do the EGFP+ and tdTomato+ cells have noticeable differential gene expression? The inclusion of RT-PCR analysis of all genes analyzed for both cell populations would bolster statements on lines 230-231 and 254-256.

We agree the reviewer’s comment and we have revised the RT-PCR panel in this revised version (Figure S4).

- Figure 4D-D', Please include cross-section images to indicate the focal plane for analysis.

We included cross-section images in this revised version (Figure 4E-E”).

- Figure 5B: Complimentary images visualizing the reduction of Notch would be helpful.

We are sorry not to include the data. In this revised version, we included notch reporter expression data that comparing WT, *Tg(krt4:il17rd)*, and *Tg(krt4:il17ra1a)* in Figure S5E.

- Line 432-433: "Moreover, we have demonstrated that IL-17 can influence cytoneme extension by regulating Cdc42 GTPases, ultimately affecting actin polymerization." This claim would be strengthened by assaying for Cdc42 activity.

It is a great idea, and we were trying to address this issue. However, we realized that activity measure with biosensors, especially in vivo, required significant amount of time and effort and validations which seem to take a substantial amount of work needed, and no confidence to work in our end. And, it seems the current methods works for in vitro samples still has many limitations such as sensitivity issues. Although, we agree cdc42 activity measure will bolster our findings, it seems very challenging to apply it to zebrafish in vivo system.

- Line 445-447: "Clint1(Clathrin Interactor 1) plays an important role in vesicle trafficking, and it is well established that endocytic pathways are critical for multiple steps in cytoneme-mediated morphogen delivery (Kalthoff et al., 2002)." Please add references to the "endocytic pathways are critical for multiple steps in cytoneme-mediated morphogen delivery" portion of the sentence.

We revised the sentence. It is “well established” -> it is “suggested”, and added a reference (Daly et al., 2022).

**Reviewer #3 (Recommendations For The Authors):**
The details of the "cytoneme inhibition" experiments need to be better clarified. How long was the dox treatment? How soon did the cells start to show "disorganization"? How soon did the KC in the periderm start to show increased proliferation?

Thank you for the valuable comment and in response, we have revised the main text as follows “Although the cytoneme inhibition is evident after overnight treatment with the inducing drugs, noticeable epidermal phenotypes begin to appear after 3 days of treatment. This reflects the higher cytoneme extension frequency and their potential role during metamorphic stages, which takes a couple of weeks (Figure 1C)”